# Adverse effects of high-fat diet consumption on contractile mechanics of isolated mouse skeletal muscle are reduced when supplemented with resveratrol

Sharn P. Shelley[1] [ID] , Rob S. James[2], Steven J. Eustace[1], Mark C. Turner[3,5] [ID] , Ryan Brett[3], Emma L. J. Eyre[4] and Jason Tallis[1]

[1] *Research Centre for Physical Activity, Sport and Exercise Science, Coventry University, Coventry, UK*
[2] *Faculty of Life Sciences, University of Bradford, Bradford, UK*
[3] *Research Centre for Health and Life Sciences, Coventry University, Coventry, UK*
[4] *School of Life Sciences, Coventry University, Coventry, UK*
[5] *Institute for Cardio-Metabolic Medicine, University Hospital Coventry and Warwickshire, Coventry, UK*

Handling Editors: Karyn Hamilton & Nathan Winn

The peer review history is available in the Supporting Information section of this article (https://doi.org/10.1113/JP287056#support-information-section).

**Abstract figure legend** An evaluation of the effectiveness of resveratrol as an anti-obesogenic nutritional strategy to mitigate the adverse effects of a high-fat diet on skeletal muscle health. Resveratrol, when consumed with a high-fat diet, reduces adipose accumulation in female CD-1 mice (∼18 weeks old), compared to a high-fat diet only. Using the work loop technique, we further show that resveratrol preserves fast-twitch extensor digitorum longus (EDL) muscle function by preventing the high-fat diet-induced decline in acute power output and cumulative work during fatiguing contractions. Notably, EDL performance in the high-fat diet + resveratrol group was comparable to that of standard laboratory diet-fed mice. Resveratrol did not alter skeletal muscle morphology or contractility when consumed with a standard laboratory diet. Despite previous reports linking resveratrol's anti-obesogenic effects to increased expression of silent information regulator 2 mammalian ortholog (SIRT1), we found no difference in SIRT1 protein expression in isolated soleus and EDL muscles across all groups. These findings suggest that resveratrol could be an effective nutritional strategy in negating some of the detrimental effects of high-fat diet consumption on skeletal muscle health.

**Abstract** Increasing evidence indicates resveratrol (RES) supplementation evokes anti-obesogenic responses that could mitigate obesity-induced reductions in skeletal muscle (SkM) contractility. Contractile function is a key facet of SkM health that underpins whole body health. For the first time, the present study examines the effects of a high-fat diet and RES supplementation on isolated soleus (SOL) and extensor digitorum longus (EDL) contractile function. Female CD-1 mice, ∼6 weeks old ($n = 38$), consumed a standard laboratory diet (SLD) or a high-fat diet (HFD), with or without RES (4 g kg$^{-1}$ diet) for 12 weeks. SOL and EDL ($n = 8$–10 per muscle, per group) were isolated and then absolute and normalised (to muscle size and body mass) isometric force and work loop power output (PO) were measured, and fatigue resistance was determined. Furthermore, sirtuin-1 expression was determined to provide mechanistic insight into any potential contractile changes. For SOL absolute force was higher in HFDRES compared to HFD ($P = 0.033$), and PO normalised to body mass and cumulative work during fatigue were reduced in HFD groups ($P < 0.014$). EDL absolute and normalised PO and cumulative work during fatigue were lower in HFD compared to other groups ($P < 0.019$). RES negated most adverse effects of HFD consumption on EDL contractility, with HFDRES producing PO and cumulative work comparable to the SLD groups. Sirtuin-1 expression was not influenced by diet in either muscle ($P > 0.165$). This study uniquely demonstrates that RES attenuates HFD-induced reductions in contractile performance of EDL, but this response is not explained by altered sirtuin-1 expression. These results suggest RES may be an appropriate strategy to alleviate obesity-induced declines in SkM function.

(Received 4 June 2024; accepted after revision 27 March 2025; first published online 10 May 2025)

**Corresponding author** S. P. Shelley and Jason Tallis: Research Centre for Physical Activity, Sport and Exercise Science, Coventry University, Coventry, CV1 5FB, UK. Email: shelley3@uni.coventry.ac.uk; ab0289@coventry.ac.uk

### Key points

- Skeletal muscle health, a precursor for disease prevention, whole body health and quality of life, is substantially reduced because of obesity.
- Growing evidence suggests that the anti-obesogenic effects of nutritional supplement resveratrol may mitigate against obesity-induced muscle pathology. However, the effect of resveratrol on skeletal muscle contractile performance, a primary marker of skeletal muscle health, is yet to be examined.
- Our findings indicate that resveratrol reduces the adverse effects of high-fat diet consumption on the contractile performance of isolated fast twitch muscle and reduces the accumulation of central adipose.
- Resveratrol had little effect on skeletal muscle performance of standard diet mice, highlighting its specific efficacy in addressing high-fat diet-induced muscle pathology.

## Introduction

Obesity is a global epidemic with profound individual, economic and societal impact. Extensive evidence links obesity to reduced skeletal muscle (SkM) health, where excessive body fat has been shown to impair SkM metabolism, regeneration and contractile function (Akhmedov & Berdeaux, 2013; Seebacher et al., 2017;

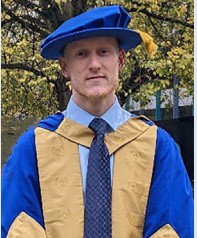

**Sharn Shelley** is currently a postdoctoral research fellow in skeletal muscle mechanics at Coventry University. During his undergraduate degree, he studied Sports and Exercise Science where he focused on biomechanics and skeletal muscle physiology, which led to him pursuing a PhD in isolated skeletal muscle mechanics at Coventry University under the guidance of Dr Jason Tallis and Professor Rob James. His current research explores the potential anti-obesogenic effects of nutraceuticals, with a focus on their potential to alleviate the adverse effects of high-fat diet consumption on the contractile performance of isolated skeletal muscle.

Tallis et al., 2017). SkM health is important to whole body health, and diminished function contributes to physical disability, morbidity, mortality and obesity (Frontera & Ochala, 2015; Morgan et al., 2020; Wu et al., 2017). As such, developing an understanding efficacious therapeutic strategies that improve SkM health in the obese phenotype is essential for improved population health, where, in this context, nutraceuticals are gaining substantial attention for potential anti-obesogenic effects.

SkM contractile function is a primary marker of SkM health because it reflects the optimisation of multiple regulatory physiological processes. Studies employing assessments of isolated SkM mechanical function in high-fat diet (HFD) models have been essential to the current understanding of obesity effects on SkM health (Tallis et al., 2018, 2021). Such approaches provide unique opportunity to more accurately determine muscle quality (function relevant to tissue size), measure direct muscle and contractile mode specific effects, and more precisely understand the influence of effect moderators such as age, duration of obesity and dietary composition, providing a level of insight unattainable in whole body *in vivo* models (Tallis et al., 2018, 2021). Although effects have been shown to be both muscle and contractile mode specific, such work has demonstrated that dietary-induced obesity can cause reduced maximal force and power relative to both body mass and muscle size, as well as impaired fatigue resistance (Ciapaite et al., 2015; Eshima et al., 2017; Hill et al., 2019; Hurst et al., 2019; Shelley et al., 2023; Tallis et al., 2017, 2024). These effects have been attributed to several interacting systemic and direct mechanisms, which are probably also muscle specific. In particular, obesity causes low grade chronic inflammation (Erskine et al., 2017) and, in combination with muscular lipid accumulation (Messa et al., 2020), this can result in reduced silent information regulator 2 mammalian ortholog (SIRT1) and $5'$-adenosine monophosphate-activated protein kinase (AMPK) activity (Tallis et al., 2017, 2018), as well as impaired SkM protein synthesis (Akhmedov & Berdeaux, 2013), mitochondrial function (Heo et al., 2017) and excitation contraction coupling (Eshima et al., 2020). This depth of understanding and the unique benefits of such experimental approaches adequality positions isolated SkM models as an important approach to understand the anti-obesogenic potential of nutraceuticals (Nishikawa, Monroy et al., 2018; Tallis et al., 2015, 2018).

Resveratrol (3,5,4$'$-trihydroxystilbene), a natural polyphenolic compound that can be sourced in trace amounts in grapes and peanuts, may be an effective nutritional strategy to promote SkM health in individuals with obesity (Tallis et al., 2021). Mechanistically RES has been shown to evoke responses opposing the obesity-induced reduction in SkM health. Specifically, evidence indicates that RES can act through cellular mechanisms that regulate SkM

oxidative metabolism and proteostasis (Dirks Naylor, 2009). A RES-induced reduction in regulation of SkM proteostasis and an inhibition of protein catabolism in particular have been attributed to attenuation of NF-$\kappa$B and IKK$\beta$ signalling (Wyke et al., 2004). *In vitro* experiments using C2C12 myoblasts have demonstrated that RES can inhibit myotoxic induced cell death and enhance myogenic differentiation through SIRT1 (Saini et al., 2012). RES has been shown to regulate glucose and lipid homeostasis via insulin-dependent and independent signalling pathways (Park et al., 2007). The beneficial effects of RES on SkM health may be most evident when cellular metabolism is compromised (Toniolo et al., 2023) and there is evidence supporting the translation of these results to both animal models and human obesity.

In humans living with overweight and obesity, RES supplementation has been demonstrated to reduce markers of inflammation, activate AMPK, and increase SIRT1 and peroxisome proliferator-activated receptor gamma coactivator 1$\alpha$ protein levels, leading to SkM mitochondrial biogenesis, enhanced mitochondrial function, and reduced intramyocellular lipids, adipocyte size and whole body adiposity (De Ligt et al., 2018; Konings et al., 2014; Méndez-del Villar et al., 2014; Polanen et al., 2021; Timmers et al., 2011). However, a paucity of research evidence and conflicting findings currently limit confidence in the anti-obesogenic potential of RES (Poulsen et al., 2013; van der Made et al., 2015). Although human trials are clearly valuable, equivocal findings are influenced by participant homogeneity, differences in trial design and dose-dependent effects that influence the molecular pathways targeted by RES (Toniolo et al., 2023). Harnessing the benefits of animal models to understand the anti-obesogenic potential of RES is important for developing an understanding of mechanistic effects, the influence of response moderators and the design of translational studies. Current findings from animal models indicate that RES directly targets adipose tissue and lipid metabolism of the liver and SkM, inducing decreased adipogenesis, increased apoptosis, and stimulation of the lipolytic and oxidative pathways, culminating in reduced adiposity in HFD mice (Aguirre et al., 2014; Kim et al., 2011; Lagouge et al., 2006; Wang et al., 2015). In rodent SkM, RES has been shown to increase AMPK and SIRT1 activity, increase mitochondrial function and lipid metabolism, and reduce chronic inflammation and intramyocellular lipids in HFD rodents (Baur et al., 2006; Kim et al., 2011; Lagouge et al., 2006; Shabani et al., 2020; Wang et al., 2015). However, whether these effects are in part attributable to activation of SIRT1 remains inconclusive (Brenner, 2022; Denu, 2005).

Although these findings infer mechanistic promise, there is a dearth of evidence evaluating the potential of RES for attenuating HFD-induced changes in contra-

ctile function. In both HFD and standard diet rodents, RES has been shown to increase *in vivo* absolute and relative forelimb grip strength, as well as fatigue resistance (Dolinsky et al., 2012; Huang et al., 2019; Kan et al., 2018; Lagouge et al., 2006; Wu et al., 2013; Zhou et al., 2019). However, such approaches are limited in elucidating effects on muscle quality, contractile mode, muscle-specific effects and the direct impact of muscle fatigue (Tallis et al., 2018, 2021). Application of an isolated muscle model to evaluate contractility addresses these limitations, providing a level of insight that cannot be ascertained with *in vivo* assessment (Tallis et al., 2018, 2021). As such, the present study harnessed the benefits of an isolated SkM model to uniquely examine the effects of RES directly on HFD-induced changes in SkM contractile function. Specifically, the study utilised the work loop (WL) model to provide a closer approximation of dynamic *in vivo* power producing SkM activity (Caiozzo, 2002; James et al., 1995; Josephson, 1993). The WL model overcomes some of the criticisms of both isometric and force-velocity experiments in evaluating muscle function, which fail to reflect the dynamic nature of muscle function and, in the case of the latter, overestimate power producing ability (James et al., 1996). As per *in vivo* dynamic muscle activity, WL assessment reflects the ability of the muscle to produce work during shortening, as well as the work absorbed during lengthening, and is influenced by the rate of muscle activation and relaxation (Josephson, 1993). This can be assessed using *in vivo* relevant stimulation parameters, length change waveforms and velocities (James et al., 1995, 1996) allowing the power output (PO) cycle frequency relationship and the response to fatiguing dynamic contractions to be determined (James et al., 1995, 1996; Josephson, 1993). Importantly, the combination of isometric and WL assessments has been integral to developing the understanding of the contractile and mode specific effects of HFD on SkM function, making this approach well positioned for determining the anti-obesogenic effects of RES (Hill et al., 2019; Hurst et al., 2019; Shelley et al., 2023; Tallis et al., 2024).

Specifically, the present study uniquely examined the effects of 12 weeks of RES and a HFD on the maximal isometric force, WL, PO and fatigue resistance of soleus (SOL; predominantly slow-twitch) and extensor digitorum longus (EDL; predominantly fast-twitch) muscle, isolated from young adult female CD-1 mice. Furthermore, given the proposed RES-induced activation of SIRT1, a commonly cited mechanism responsible for offsetting the adverse effects of HFD on metabolic health (Lagouge et al., 2006), the present study determined SIRT1 expression aiming to explore its potential role in HFD or RES-induced changes in isolated SkM contractility. It was hypothesised that the magnitude of body mass and adipose accumulation, as well as the reduction in contractile performance evoked via HFD, would be reduced by RES consumption, which in part would be attributed to RES increasing SIRT1 expression.

## Methods

### Ethical approval

Institutional (P108131) and UK Home Office approval (PP4247175) were granted for the use of animals and the methods used in this project. All procedures were carried out in accordance with the British Home Office Animals (scientific procedures) Act 1986. The present study complies with the ethical principles under which *The Journal of Physiology* operates.

### Animals

Female CD-1 mice, purchased at ~4 weeks old (Charles River, Margate, UK), were housed in groups of five at British Home Office approved facility and kept under 12:12 h light/dark photocycles at 22°C and 50% relative humidity in individually vented cages. At ~4 weeks of age, mice were randomly split into four experimental groups (total starting sample: $n = 10$ per group; final sample $n = 10$ SLDRES and HFDRES; $n = 9$ SLD and HFD) and, following 13 days of habituation, which included gradual transition from standard lab chow (TestDiet 5755; Purinina TestDiet, Richmond, IN, USA; calories provided by protein 18.3%, fat 22.1% and carbohydrate 59.6%; gross energy 4.07 kcal $g^{-1}$] to new respective custom diets, mice consumed one of the following for 12 weeks: (1) standard low-fat diet (TestDiet 58Y2; calories provided by protein 18.0%, fat 10.2% and carbohydrate 71.8%; gross energy 3.76 kcal $g^{-1}$) (SLD); (2) high-fat diet (TestDiet 58V8; calories provided by protein 18.1%, fat 46.2% and carbohydrate 35.7%; gross energy 4.62 kcal $g^{-1}$) (HFD); (3) SLD enriched with RES (4 g $kg^{-1}$ feed); and (4) HFD enriched with RES (4 g $kg^{-1}$ feed). Animal health was monitored weekly using an Animal Welfare and Ethical Review Body (i.e. AWERB) approved checklist, which included monitoring body mass weekly (Fig. 2). Mice were provided with *ad libitum* access to food and water throughout. Procedures were carried out blinded.

A 12 week HFD was used as identical and similar durations have previously demonstrated changes in body composition and *in vitro* muscle mechanics (Bott et al., 2017; Eshima et al., 2017; Matsakas et al., 2015; Tallis et al., 2017). The RES dosing strategy implemented in this study (4 g $kg^{-1}$ diet for 12 weeks) was selected based on previous evidence indicating that an identical dose, over similar feeding durations (10–15 weeks), elicits: (1) a reduction in adiposity in HFD rodents compared to SLD controls (Lagouge et al., 2006; Mendes et al., 2016; Shabani et al., 2020); (2) improved voluntary *in vivo* fatigue resistance in

SLD and HFD mice (Lagouge et al., 2006) and SLD rats (Dolinsky et al., 2012); and (3) improved *in vivo* relative grip strength in HFD mice compared to HFD controls (Lagouge et al., 2006).

### Muscle preparations

Animals were culled via cervical dislocation (in accordance with the British Home Office Animals Scientific Procedures Act 1986, Schedule 1) following the 12 week dietary intervention. From the point of the death of the animal to final contractile assessment, the entire protocol lasted ∼210 min. Animals were weighed to determine body mass and nasoanal length measured using digital callipers (Fisher Scientific 3417; Fisher Scientific, Loughborough, UK), from which body mass index (BMI) and Lee index of obesity (Bernardis & Patterson, 1968) were calculated.

$$\text{BMI} = \frac{BM \ (\text{g})}{\left[ NAL \ (\text{cm})^2 \right]} \div 100 \qquad (1)$$

where eqn (1) is the rodent body mass index calculation (Sjögren et al., 2001).

$$\text{Lee index of obesity} = \frac{\sqrt[3]{BM \ (\text{g})}}{NAL \ (\text{cm})} \times 1000 \qquad (2)$$

where eqn (2) is the Lee index of obesity calculation (Bernardis & Patterson, 1968).

The s.c. fat pad around the top of the hindlimbs and genitals was extracted and weighed as an indication of overall adiposity (Rogers & Webb, 1980). The ratio of body mass:fat mass was determined as a measure of relative adiposity (Hill et al., 2019). In addition, whole EDL and SOL ($n = 8$–10 per muscle per group) were dissected from the right and left hind limbs respectively. Muscle isolations were conducted in refrigerated (1–3°C) oxygenated (95% $O_2$, 5% $CO_2$) Krebs Henseleit solution (in mM: 118 NaCl; 4.75 KCl; 1.18 $MgSO_4$; 24.8 $NaHCO_3$; 1.18 $KH_2PO_4$; 10 glucose; 2.54 $CaCl_2$ in each case; pH 7.55 at room temperature before oxygenation; pH 7.23–7.27 at 1–3°C oxygenated; pH 7.4–7.42 at 37°C oxygenated). The SOL (68.3% type I; 31.7% type IIa at 20 weeks for female CD-1 mice; Messa et al., 2020) and EDL (EDL; 2.6% type I; 22.3% type IIa; 26.7% type IIx; 46.3% IIb, 2.1% type IIxb at 20 weeks for female CD-1 mice; Messa et al., 2020) represent locomotor muscles that differ in fibre type and function, allowing understanding of muscle and fibre type-specific effects. The tendon attachment at the proximal end of the tissue preparation was left intact with a small piece of bone still attached. The distal tendons were secured with an aluminium foil T-clip as an anchoring point for attachment and to avoid tendon slippage during

muscle force production (Ford et al., 1977; Goldman & Simmons, 1984; James et al., 2005).

### Experimental equipment

Muscles were placed in a perspex flow through chamber filled with circulated constantly oxygenated (95% $O_2$, 5% $CO_2$) Krebs Henseleit solution. The reservoirs of Krebs solution were stored in external heater/cooler bath (Grant LTD6G; Grant Instruments, Shepreth, UK), which were adjusted to maintain a physiologically relevant temperature of $37 \pm 0.2$°C inside the muscle bath. Bath temperature was continuously monitored using a digital thermometer (Traceable, Fisherbrand; Fisher Scientific). Using the bone at the proximal end and T-foil clip at the distal end, the muscle was attached to crocodile clips connected to a force transducer (UF1; Pioden Controls Ltd, Ashford, UK) at one end and motor arm (V201; Ling Dynamic Systems, Royston, UK) at the other. Muscle activation occurred via external electric stimulation being applied to platinum electrodes positioned parallel to the muscle within the bath using a tabletop power supply (PL320; Thurlby Thandar Instruments, Huntington, UK). Stimulation and length change parameters were controlled through custom-written software (Testpoint, CEC; Measurement Computing, Norton, MA, USA) on a standard desktop personal computer, via a D/A board (KPCI3108; Keithley Instruments, Cleveland, OH, USA). Visualisation of changes in force and length was provided by a storage oscilloscope (2211 or 1002; Tektronix, Marlow, UK). Motor arm position was detected using a Linear Variable Displacement Transformer (DFG5.0; Solartron Metrology, Bognor Regis, UK). All contractile data were sampled at 10 kHz.

### Isometric force

The methods employed for evaluating isometric properties, WL PO and fatigue of isolated mouse EDL and SOL in this study are based on established procedures. The protocol and experimental equipment used have been described in detail previously (Hill et al., 2019; Shelley et al., 2022, 2023; Tallis et al., 2017). As such, a brief description of all contractile procedures is provided below. Following a 10 min acclimatisation period, each muscle underwent a series of isometric twitch activations (Stimulation current: 160 mA; pulse width: 2 ms) to determine the optimal length and stimulation voltage (typically 12–16 V for SOL and 14–18 V for EDL) required for maximal isometric twitch force. Muscle length optimal for twitch performance (defined as $L_0$), determined using an eyepiece graticule and microscope, was utilised as the starting length for all subsequent contractile assessments. The estimated fibre lengths

for SOL and EDL were calculated as 85% and 75% of $L_0$, respectively, in accordance with previous research (James et al., 1995). Using $L_0$ and stimulation parameters for twitch performance, measures of tetanic force were recorded using a fixed burst duration of at 350 ms for SOL and 250 ms for EDL. Stimulation frequency was manipulated until maximal tetanic force plateaued (typically 120 Hz and 230 Hz for the SOL and EDL respectively). After maximal tetanic force had been achieved, a control tetanus was performed at the first stimulation frequency to monitor change in contractile performance over time. From the maximal tetanus response, time to half peak tetanus force (THPT) and last stimulus to half relaxation (LSHR) were recorded as indicative measures of activation and relaxation, as well as $Ca^{2+}$ kinetics (Ebashi & Endo, 1968). A 5 min rest was implemented after each tetanic or WL activation to allow for metabolic recovery (Altringham & Young, 1991; Askew & Marsh, 1997)

### WL PO

The WL technique was utilised to examine the contractile performance of isolated muscle during cyclical length changes, considered to be the *in vitro* model that comes closest to emulating *in vivo* dynamic muscle activity (James et al., 1995, 1996; Josephson, 1985, 1993). Each muscle was subjected to four sinusoidal length changes around the muscles resting length. Because muscle activation is not instantaneous, muscle-specific phasing (typically –2 ms and –10 ms for the SOL and EDL repetitively) of electrical stimulation was provided prior to the muscle reaching its greatest length in the WL cycle so that work was maximised during shortening. Stimulation frequency was increased from that used for tetanic activations to 160 Hz and 260 Hz for the SOL and EDL, respectively, because previous work indicates that these stimulation frequencies evoke maximal WL PO (Shelley et al., 2022; Vassilakos et al., 2009). A range of cycle frequencies (CF; rate at which each muscle undergoes sinusoidal length change cycle, i.e. analogous to shortening velocity) were utilised to examine whether the CF that elicited maximal PO was influenced by a HFD or RES. Initially, CFs of 5 Hz and 10 Hz were used to examine the SOL and EDL, respectively, because previous work has established that these muscles typically achieve maximal WL PO using these parameters (James et al., 1995; Tallis et al., 2017). Once parameters for optimal PO at 5 Hz and 10 Hz CF for the SOL and EDL, respectively, were identified, these CFs were then used as the control WLs for the remainder of the experiment. Additionally, passive net work was recorded during unstimulated WLs at the control CFs to determine

the contribution of passive net work to maximal active net work. Once maximal WL PO and passive net work were determined at the control CFs, PO was then assessed across a range of CFs, in a randomised order, to establish a PO–CF curve for each experimental group (Hill et al., 2019; Hurst et al., 2019). Burst duration (duration of electrical stimuli), strain (symmetrical length change around starting length, e.g. 0.1 strain results in the muscle lengthening by 5%, shortening by 10% before being re-lengthened by 5% back to starting length) and phasing parameters were adjusted to optimise net work produced during the WL cycle at each CF. Typical parameters utilised to elicit maximal net work of each muscle at each CF are detailed previously (Shelley et al., 2023). During each assessment, muscles were subjected to four WL cycles and data were used from the WL that produced peak power. During the length change cycle, force and length were plotted against each other forming a WL (WL example is provided in Fig. 1, which highlights key aspects of a WL, including burst duration, strain and phasing). Instantaneous power (instantaneous velocity × instantaneous force) was calculated at each time point during the WL and was averaged over the entire WL to determine net PO. After every three CFs, and prior to fatigue assessments, PO was determined using the initial parameters to assess deterioration of PO over time; isolated muscle contractile performance will deteriorate over time as a result of the build-up of an anoxic core (Barclay, 2005). Assessing PO with set parameters over the course of the experiment allowed for the correction of net work for other CFs relative to the control WLs (Hill et al., 2019; Hurst et al., 2019; Vassilakos et al., 2009). Muscle performance prior to the fatigue run declined by $6 \pm 6\%$ over a time period of $\sim$180 min, similar to studies utilising a similar methodological approach (Hill et al., 2019; James et al., 1996). Once the final control WL cycles were performed, the muscle rested for 10 min prior to assessment of fatigue and recovery.

### Fatigue resistance, cumulative work and recovery

The fatigue protocol consisted of each muscle undergoing 50 consecutive WL cycles using the initial CFs and control parameters, except for stimulation frequency, which was reduced to 100 Hz and 200 Hz for the SOL and EDL, respectively. Reducing stimulation frequency for assessments of fatigue has been shown to better represent the *in vivo* mechanism of fatigue (Shelley et al., 2022; Vassilakos et al., 2009). The net work of every loop was recorded and plotted relative to maximal PO recorded during the fatigue protocol. Both time taken for power to fall below 50% maximal power output and total cumulative work production (sum of net work performed

in each cycle; Askew et al., 1997) were recorded to infer the relative and absolute differences in fatigue resistance (Shelley et al., 2023). Recording relative and absolute differences accounts for potential differences in absolute PO between experimental groups, which may promote a faster rate of fatigue. Every 10 min for 30 min post fatigue run, each muscle was subjected to one set of WL cycles using the same contractile parameters for the control WLs to assess recovery. Net work during recovery was expressed as a percentage of pre-fatigue maximal PO.

### Muscle mass and dimensions calculations

Post contractile assessments each preparation was removed from the muscle chamber. Following the removal of tendons, T-foil clip and bone, the muscle tissue only was blotted on absorbent paper to remove the excess Krebs solution and weighed to determine wet muscle mass (TL-64; Denver Instrument Company, Arvada, CO, USA). Mean muscle cross-sectional area (CSA) was calculated from $L_0$, muscle mass and an assumed density of 1060 kg m$^{-3}$ (Brooks & Faulkner, 1991; Méndez & Keys, 1960). Isometric stress (kN m$^{-2}$) was calculated by dividing maximal tetanic force by CSA. Absolute WL PO was divided body mass and muscle mass to determine WL PO normalised to body mass (W kg$^{-1}$ body mass) and WL PO normalised to muscle mass (W kg$^{-1}$ muscle mass), with the later comprising a marker of muscle quality.

### Protein extraction, quantification and immunoblotting

SOL and EDL SkM tissue was homogenised in cold RIPA lysis buffer containing protease and phosphatase inhibitor cocktail (Fisher Scientific) and centrifuged at 12,000 *g* for 10 min to pellet insoluble material. Protein concentrations were determined using a Pierce BCA protein assay in accordance with the manufacturer's instructions (Fisher Scientific). Samples were mixed with dH2O, 4× sample buffer (Bio-Rad, Hercules, CA, USA) containing $\beta$-mercaptoethanol (Sigma, St Louis, MO, USA) and heated at 95°C for 5 min. A protein concentration of 25 µg was loaded on to 4%–20% mini-TGX (Bio-Rad) and separated by electrophoresis. Proteins were transferred onto poly(vinylidene difluoride) membrane (Bio-Rad) before being blocked in 5% non-fat milk (Marvel; Premier Foods Group, St Albans, UK) for 1 h at room temperature. Membranes were incubated overnight with appropriate primary antibody at 4°C. The antibodies were sirtuin 1 (SIRT1) (#07-131) and glyceraldehyde-3-phosphate dehydrogenase (GAPDH) (D16H11) (#5174) in SuperBlock blocking buffer (#37 535) or 5% milk at a concentration of 1:1000 and 1:5000, respectively. Membranes were washed in Tris-buffered saline with Tween 20, incubated with horse-radish peroxidase conjugated goat, anti-rabbit secondary antibody (#12-348) at 1:10 000 and incubated with SuperSignal West Femto chemiluminescent substrate (Fisher Scientific) for 5 min. Bands were visualised and densities determined using a Odessey Fc Imaging System

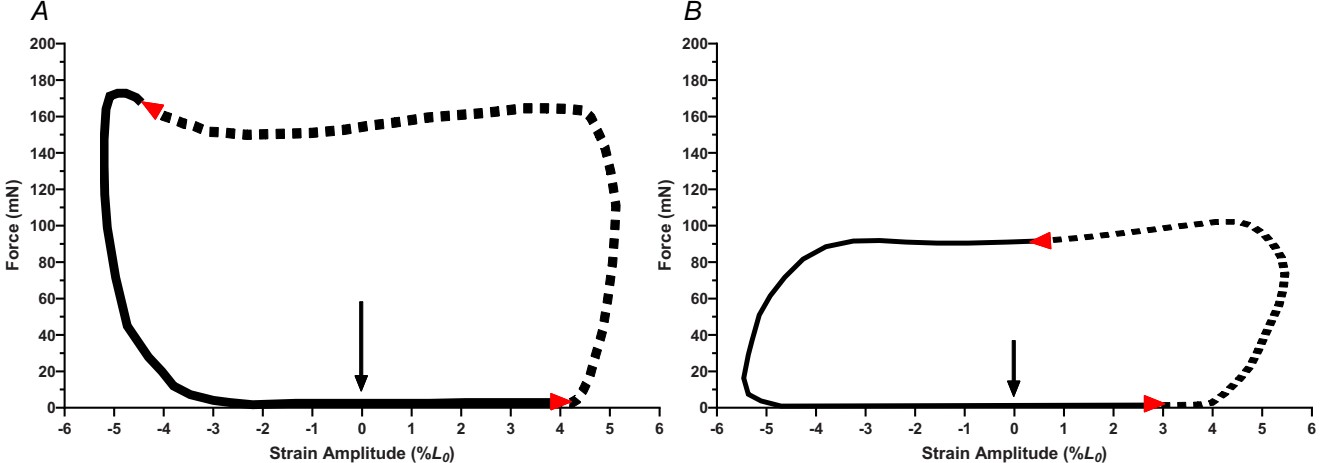

**Figure 1. Example Extensor digitorum longus (A) and soleus (B) work loops**
Extensor digitorum longus (*A*) and soleus (*B*) work loop cycle optimised for maximal work at 10 Hz and 4 Hz cycle frequency and 260 Hz and 160 Hz stimulation frequency, respectively. Muscle and cycle frequency specific strain amplitude (EDL: 5%; SOL: 5.5%), burst duration (EDL: 50 ms; SOL: 92 ms) and phasing (EDL: −2 ms; SOL: −10 ms) were used to achieve maximal work. All work loops proceed in an anti-clockwise direction (denoted via red arrows), with the initiation of all work loops starting at the muscles resting length ($L_0$) as indicated via the black arrow. The dotted line in each loop represents the portion of the WL that is electrically stimulated (i.e. burst duration). Adapted from Shelley et al. (2024). [Colour figure can be viewed at wileyonlinelibrary.com]

(LI-COR, Lincoln, NE, USA) and Image Studio, version 5.2.5 (LI-COR).

## Statistical analysis

Statistical analysis was performed using SPSS, version 28.0 (IBM Corp., Armonk, NY, USA) and Prism, version 10.2.3 (GraphPad Software Inc., San Diego, CA, USA). All data are presented as the mean ± SD. Following checks of normality and homogeneity of variance (Levenes), parametric analysis was performed. Mixed model repeated measures analysis of variance [ANOVA; with HFD, RES treatment (between subjects) and weeks (within subjects) used as the factors] was utilised to determine the effect of HFD and RES consumption on weekly body mass. Two factor ANOVA was conducted, with HFD (SLD *vs.* HFD) and treatment (Control *vs.* RES) as the fixed factors, to examine whether differences existed in measures of animal and muscle morphology (body mass, muscle mass, muscle length and muscle CSA), isometric properties (absolute tetanic force, tetanic stress, THPT and LSHR), passive net work, passive net work as a percentage of maximal active net work, time to reach 50% of maximal PO during fatigue protocol, cumulative work produced after 50 consecutive WLs and SIRT1:GADPH expression. Mixed model repeated measures ANOVA, with HFD, RES (between subjects) and cycle frequency (within subjects) used as the factors, was utilised to determine any changes in SOL and EDL absolute PO and PO normalised to body mass and muscle mass. Mixed model repeated measures ANOVA, with HFD, RES treatment (between subjects) and stimulation frequency (within subjects) used as the factors, was utilised to determine whether the reduction in stimulation frequency used for assessment of fatigue resulted in any significant changes in SOL and EDL PO normalised to muscle mass and, if so, whether the changes in PO were uniform across diet and treatment groups. Mixed model repeated measures ANOVA, with HFD, RES treatment (between subjects) and time (within subjects) as the factors, was utilised to examine the recovery of WL PO following the fatigue protocol. Significant interactions observed for ANOVA were explored using Bonferroni *post hoc* for multiple comparisons. Partial eta squared ($\eta p^2$) was calculated to estimate effect sizes for all significant main effects. Thresholds for partial eta squared effect size were classified as small (<0.05), moderate (0.06–0.137) or large (>0.138) (Cohen, 1988). Where appropriate, Cohen's *d* was calculated to measure effect size of interactions observed and was then corrected for bias using Hedges *g* as a result of the appropriate sample size (Hedges, 1981). Cohen's *d* effect size was interpreted as trivial (<0.2), small (0.2–0.6), moderate (0.6–1.2) or large (>1.2) (Hopkins et al., 2009). $P < 0.05$ was considered statistically

significant. Given the volume of statistical analysis undertaken, exact *P* values are presented and summarised in the Results. The results of *post hoc* tests specific to evaluating the effects of cycle frequency are only summarised in text.

The SPM1D package (Todd Pataky, version M 0.1; https://spm1d.org) in MATLAB R2018b (MathWorks Inc., Natick, MA, USA) was used to perform statistical parametric mapping (SPM) on fatigue data (cumulative work production and percentage decline in PO) (Pataky, 2010). SPM calculates the *F* (ANOVA) or *t* (*t* test) value on every data point obtained during the fatigue protocol; instead of calculating a *P* value for every data point, inferential statistics are based on random field theory and thus maintain a constant error of $\alpha$ (Pataky et al., 2013). First, one-way ANOVA SPM[*F*] statistics was used to determine whether there was a main effect of group (SLD, SLDRES, HFD, HFDRES) on cumulative work and percentage decline in fatigue relative to maximum PO. Where clusters crossed the critical threshold, a *P* value was calculated (Pataky et al., 2013). Where main effects were observed, *post hoc* two-sample SPM[*t*] (two-sided *t* test) was conducted between each group (SLD, SLDRES, HFD, HFDRES) separately to identify which groups differed and where the differences occurred (Pataky et al., 2015). Where clusters crossed the critical threshold, this indicated a significant difference at $P < 0.05$.

## Results

### Morphology

There was a significant time × HFD interaction observed for weekly body mass ($P < 0.001$, $\eta p^2 = 0.537$) (Fig. 2). Bonferroni multiple comparisons revealed that HFD groups had significantly greater body mass from week 2 onwards compared to SLD groups ($P < 0.002$).

Final whole animal body mass, body length, BMI and Lee index of obesity were significantly higher in HFD groups compared to SLD groups ($P < 0.003$, $\eta p^2 > 0.232$) (Table 1). For fat mass and adiposity:body mass, a significant HFD × RES interaction was observed ($P < 0.038$, $\eta p^2 > 0.121$). Bonferroni multiple comparisons indicate that fat mass and adiposity:body mass was significantly higher in HFD compared to SLD ($P < 0.001$, $d = 2.67$ and $P < 0.001$, $d = 2.61$, respectively), HFD compared to HFDRES ($P = 0.011$, $d = 1.03$ and $P = 0.007$, $d = 1.14$, respectively) and HFDRES compared to SLDRES ($P < 0.001$, $d = 2.33$ and $P = 0.002$, $d = 2.18$, respectively), but no differences between SLD groups were observed ($P = 0.720$, $d = 0.28$ and $P = 0.704$, $d = 0.26$, respectively). No significant differences were observed for EDL or SOL muscle mass, fibre length or CSA ($P > 0.066$, $\eta p^2 < 0.099$).

**Table 1. Whole body and muscle specific animal morphology.**

| | SLD | SLD-RES | HFD | HFD-RES | HFD effect | | RES effect | | Interaction | |
|---|---|---|---|---|---|---|---|---|---|---|
| | (*n* = 9) | (*n* = 10) | (*n* = 9) | (*n* = 10) | *P* value | $\eta p^2$ | *P* value | $\eta p^2$ | *P* value | $\eta p^2$ |
| ***Whole body*** | | | | | | | | | | |
| Body mass (g) | 35.0 ± 4.7 | 36.7 ± 4.8 | 48.8 ± 4.4 | 47.8 ± 6.2 | **<0.001** | **0.599** | 0.843 | 0.001 | 0.443 | 0.017 |
| Body length (mm) | 101 ± 5 | 101 ± 4 | 106 ± 3 | 107 ± 3 | **<0.001** | **0.317** | 0.555 | 0.010 | 0.702 | 0.004 |
| Fat mass (g) | 1.1 ± 0.6 [a] | 1.2 ± 0.7 [a] | 4.6 ± 1.7 [b] | 3.2 ± 0.9 [c] | **<0.001** | 0.611 | 0.108 | 0.074 | **0.038** | **0.121** |
| Lee index of obesity | 325 ± 9 | 329 ± 18 | 346 ± 11 | 339 ± 15 | **0.003** | **0.232** | 0.759 | 0.003 | 0.225 | 0.043 |
| BMI (g cm$^{-2}$) | 0.34 ± 0.02 | 0.36 ± 0.05 | 0.44 ± 0.04 | 0.42 ± 0.05 | **<0.001** | **0.429** | 0.908 | <0.001 | 0.210 | 0.046 |
| Adiposity:body mass (%) | 2.8 ± 1.4 [a] | 3.2 ± 1.5 [a] | 9.3 ± 3.2 [b] | 6.5 ± 1.5 [c] | **<0.001** | 0.591 | 0.086 | 0.084 | **0.027** | **0.136** |
| ***EDL*** | *n* = 8 | *n* = 10 | *n* = 9 | *n* = 10 | | | | | | |
| Muscle mass (mg) | 10.7 ± 1.2 | 11.1 ± 1.0 | 11.3 ± 0.7 | 11.2 ± 0.6 | 0.246 | 0.041 | 0.559 | 0.010 | 0.348 | 0.027 |
| Fibre length (mm) | 9.2 ± 0.3 | 9.2 ± 0.8 | 8.9 ± 0.5 | 8.9 ± 0.4 | 0.089 | 0.089 | 0.779 | 0.002 | 0.898 | 0.001 |
| CSA (mm$^2$) | 1.10 ± 0.12 | 1.14 ± 0.15 | 1.21 ± 0.08 | 1.19 ± 0.07 | 0.066 | 0.099 | 0.737 | 0.003 | 0.459 | 0.017 |
| ***SOL*** | *n* = 8 | *n* = 10 | *n* = 8 | *n* = 10 | | | | | | |
| Muscle mass (mg) | 8.7 ± 1.4 | 8.5 ± 1.1 | 8.4 ± 0.6 | 9.2 ± 1.3 | 0.594 | 0.009 | 0.526 | 0.013 | 0.238 | 0.043 |
| Fibre length (mm) | 9.4 ± 0.5 | 9.5 ± 0.5 | 9.5 ± 0.6 | 9.4 ± 0.5 | 0.736 | 0.004 | 0.780 | 0.002 | 0.724 | 0.004 |
| CSA (mm$^2$) | 0.88 ± 0.14 | 0.84 ± 0.10 | 0.84 ± 0.06 | 0.92 ± 0.13 | 0.855 | 0.001 | 0.244 | 0.042 | 0.155 | 0.062 |

*Note*: Values are presented as the mean ± SD; bold values indicate a significant difference at *P* < 0.05. Values with different letters (*a*, *b* and *c*) are significantly different at *P* < 0.05 utilised for when significant interactions were observed.

### Isometric performance

For EDL, there were no significant HFD ($P > 0.457$, $\eta p^2 < 0.017$), RES ($P > 0.483$, $\eta p^2 < 0.017$) or HFD × RES interactions ($P > 0.091$, $\eta p^2 < 0.083$) (Fig. 3*A* and *B*) for maximal tetanus force or stress. A significant HFD × RES interaction was observed for

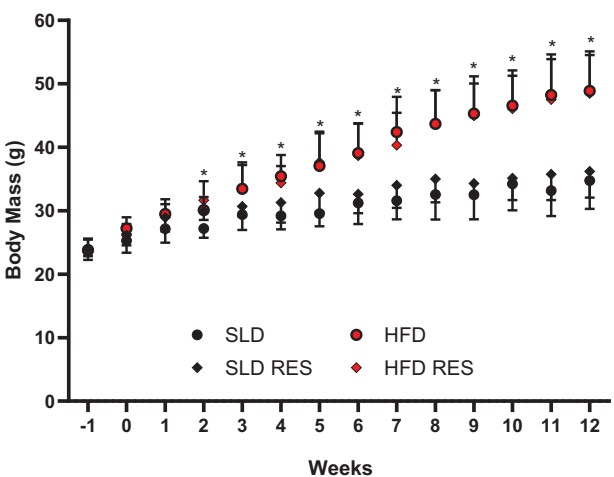

**Figure 2. Effect of 12 weeks high-fat diet and resveratrol on mouse body mass**
Effect of high-fat diet and resveratrol on body mass over 12 weeks. Data are presented as the mean ± SD with error bars in one direction only for clarity; SLD and HFD, *n* = 9; SLD RES and HFDRES, *n* = 10; *significant difference between HFD and SLD groups at *P* < 0.002; week −1 and 0 reflect habituation period where animals were gradually transitioned to their respective custom diet. [Colour figure can be viewed at wileyonlinelibrary.com]

THPT ($P = 0.033$, $\eta p^2 = 0.131$), although Bonferroni pairwise comparisons indicate no differences (SLD *vs.* HFD: $P = 0.245$, $d = 0.52$; SLD *vs.* SLDRES: $P = 0.090$, $d = 0.91$; HFD *vs.* HFDRES: $P = 0.171$, $d = 0.065$; SLDRES *vs.* HFDRES: $P = 0.052$, $d = 1.07$). LSHR time was significantly longer in HFD groups compared to SLD ($P < 0.001$, $\eta p^2 = 0.283$) (Fig. 3*D*).

For SOL, there was a significant HFD × RES interaction for absolute tetanus force ($P = 0.030$, $\eta p^2 = 0.139$) (Fig. 3*E*). Bonferroni multiple comparisons revealed that the HFDRES group produced greater (∼24%) tetanus force compared to the HFD group ($P = 0.017$, $d = 1.64$), but no other differences were observed (SLD *vs.* HFD: $P = 0.129$, $d = 0.61$ SLD *vs.* SLDRES: $P = 0.486$, $d = 0.46$; HFDRES *vs.* SLDRES: $P = 0.106$, $d = 0.83$). There were no significant HFD ($P > 0.584$, $\eta p^2 < 0.009$), RES ($P > 0.151$, $\eta p^2 < 0.063$) or HFD × RES interactions ($P > 0.160$, $\eta p^2 < 0.061$) for tetanus stress, THPT or LSHR.

### Passive work

For the EDL, there were no significant HFD ($P > 0.101$, $\eta p^2 < 0.079$), RES ($P > 0.214$, $\eta p^2 < 0.046$) or HFD × RES interactions ($P > 0.058$, $\eta p^2 < 0.105$) (Fig. 4*A* and *B*) for passive net work or passive net work as a percentage of maximal active net work at 10 Hz cycle frequency.

For the SOL, HFD groups produced greater passive net work and passive net work as a percentage of maximal active net work at 5 Hz cycle frequency ($P < 0.011$, $\eta p^2 > 0.187$) (Fig. 4*C* and *D*). There was no RES main

effect ($P > 0.145$, $\eta p^2 < 0.065$) or HFD × RES interaction ($P > 0.635$, $\eta p^2 < 0.007$).

## Maximal WL PO

For EDL maximal PO, PO normalised to muscle mass and PO normalised to body mass, there was a significant HFD × RES interaction ($P < 0.037$, $\eta p^2 > 0.125$) (Fig. 5*A*–*C*). Bonferroni pairwise comparisons revealed all outcomes were significantly lower in HFD only group compared to all other groups (maximal PO: HFD *vs.* SLD: $P = 0.036$, $d = 0.72$–$0.96$; HFD *vs.* HFDRES: $P = 0.008$, $d = 1.56$–$1.64$; PO normalised to muscle mass: HFD *vs.* SLD: $P = 0.001$, $d = 1.13$–$1.53$; HFD *vs.* HFDRES: $P < 0.001$, $d = 1.65$–$1.97$; PO normalised to body mass: HFD *vs.* SLD: $P < 0.001$, $d = 1.55$–$1.79$; HFD *vs.* HFDRES: $P = 0.019$, $d = 1.01$–$1.37$) and the other groups did not significantly differ (maximal PO: SLD *vs.* SLDRES: $P = 0.875$, $d = 0.02$–$0.32$; SLDRES *vs.* HFDRES: $P = 0.492$, $d = 0.29$–$0.66$; PO normalised to muscle mass: SLD *vs.* SLDRES: $P = 0.319$, $d = 0.02$–$0.71$; SLDRES *vs.* HFDRES: $P = 0.260$, $d = 0.32$–$0.91$; PO normalised to body mass: SLD *vs.* SLDRES: $P = 0.526$, $d = 0.06$–$0.48$; SLDRES *vs.* HFDRES: $P = 0.102$, $d = 0.55$–$0.77$). For EDL, maximal PO and PO normalised to muscle mass, there was a significant main effect of CF ($P < 0.001$, $\eta p^2 = 0.675$) (Fig. 5*A* and *B*), indicating that maximal PO occurred between 8 Hz and 12 Hz ($P > 0.999$), with the PO values from the remaining cycle frequencies being significantly lower ($P < 0.04$).

For maximal PO normalised to body mass, there was a significant HFD × CF ($P = 0.030$, $\eta p^2 = 0.080$) (Fig. 5*C*) interaction observed. Bonferroni multiple comparisons reveal that, for 12 Hz SLD groups, EDL PO was significantly greater than at 4 Hz ($P < 0.001$), whereas, for HFD groups, EDL PO was significantly higher at 12 Hz compared to 4 Hz ($P < 0.001$) and 16 Hz ($P = 0.031$).

For SOL, absolute PO and PO normalised to muscle mass, there were no significant main effects of HFD ($P > 0.262$, $\eta p^2 < 0.039$), RES ($P > 0.141$, $\eta p^2 < 0.066$) or HFD × RES interaction ($P > 0.436$, $\eta p^2 < 0.019$) (Fig. 5*D* and *E*). For PO normalised to body mass, HFD groups produced significantly lower PO compared to SLD groups ($P < 0.001$, $\eta p^2 = 0.324$) (Fig. 5*F*). There was no effect of RES ($P = 0.423$, $\eta p^2 = 0.020$) or HFD × RES interaction ($P = 0.379$, $\eta p^2 = 0.024$). For SOL, maximal PO and PO normalised to muscle mass, there was a significant main effect of CF ($P < 0.001$, $\eta p^2 = 0.675$) (Fig. 5*D* and *E*), which identified that maximal PO occurred between 3 Hz and 4 Hz ($P > 0.084$) with the PO values from the remaining cycle frequencies being significantly lower ($P < 0.04$), except between 2 Hz and 4 Hz for absolute PO, albeit the difference was approaching significance ($P = 0.055$). For maximal SOL PO normalised to body mass, there was a significant HFD × CF interaction ($P = 0.008$, $\eta p^2 = 0.139$) (Fig. 5*F*). Bonferroni pairwise comparisons indicated that, in SLD groups, PO did not differ between 2 Hz and 5 Hz ($P > 0.999$) and 3 Hz and 4 Hz ($P > 0.999$), but PO from all other cycle frequencies

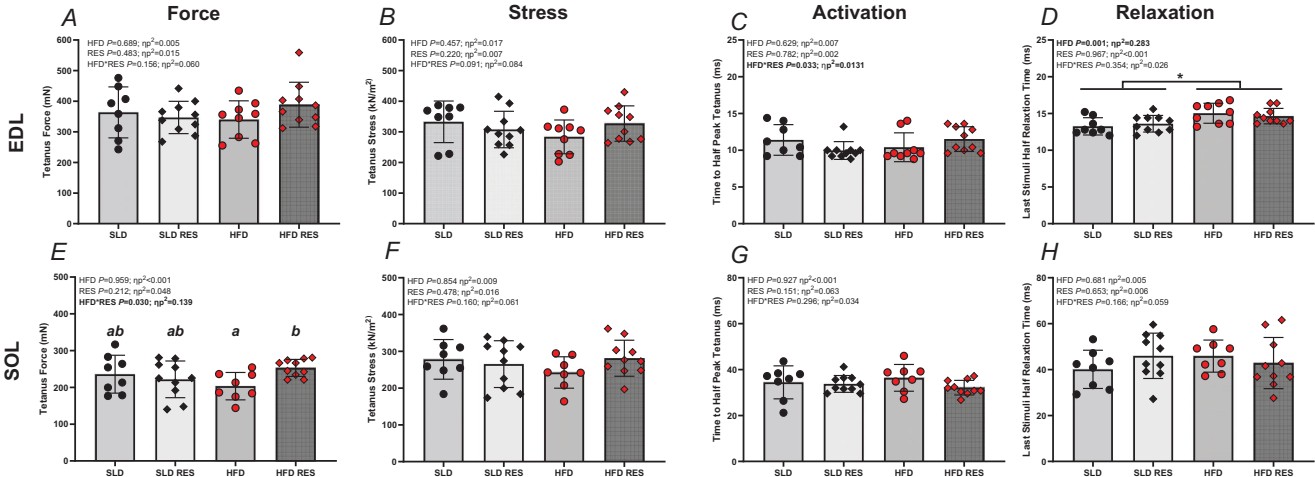

**Figure 3. The effect of 12 weeks high-fat diet and resveratrol on isolated mouse EDL and SOL isometric function**
The effect of 12 weeks of a high-fat diet and resveratrol on maximal tetanus force (*A* and *E*), maximal tetanus stress (*B* and *F*), time to half peak tetanus (*C* and *G*) and last stimuli half relaxation time (*D* and *H*) of isolated mouse EDL (*A*–*D*, respectively) and SOL (*E*–*H*, respectively). EDL, $n = 10$ for SLDRES and HFDRES, $n = 8$ for SLD and $n = 9$ for HFD; SOL, $n = 10$ for SLDRES and HFDRES and $n = 8$ for SLD and HFD. Data are presented as individual data points with bar and error lines representing the mean ± SD; *significant difference between HFD groups and SLD groups at $P = 0.001$; Different lowercase letters indicate a significant difference at $P = 0.017$ utilised for a significant interaction. [Colour figure can be viewed at wileyonlinelibrary.com]

were different ($P < 0.01$). In HFD groups, PO did not differ between 2 Hz and 4 Hz ($P > 0.999$) and 3 Hz and 4 Hz ($P = 0.088$), but PO values from all other cycle frequencies were different ($P < 0.012$).

## Fatigue and recovery

The lower stimulation frequency used for assessment of fatigue resulted in a significant reduction of maximal WL PO normalised to muscle mass of the SOL (∼25% reduction) and EDL (∼15% reduction) ($P < 0.001$, $\eta p^2 > 0.799$) (Fig. 6), but the magnitude of reduction in PO was not influenced by HFD or RES ($P > 0.211$, $\eta p^2 < 0.048$). For EDL only, there was a significant HFD × RES interaction ($P = 0.001$, $\eta p^2 = 0.292$) (Fig. 6A). Bonferroni multiple comparisons indicate that

EDL PO normalised to muscle mass was lower in HFD group compared to all other groups (HFD *vs.* SLD: $P < 0.001$, $d = 1.37$–1.53; HFD *vs.* HFDRES: $P < 0.001$, $d = 1.92$–2.58), but other groups did not differ (SLD *vs.* SLDRES: $P = 0.271$, $d = 0.28$–0.61; SLDRES *vs.* HFDRES: $P = 0.467$, $d = 0.31$–0.36) irrespective of stimulation frequency.

For the EDL, RES groups fatigued to 50% maximum PO quicker than control groups ($P = 0.045$, $\eta p^2 = 0.120$) (Fig. 7A). No significant HFD effect ($P = 0.882$, $\eta p^2 = 0.001$) or HFD × RES interaction was observed ($P = 0.196$, $\eta p^2 = 0.052$). ANOVA SPM[$F$] indicated a significant effect of experimental group on percentage decline of PO relative to maximum for the EDL (Fig. 7A $P < 0.001$). The SPM[$t$] $t$ test results indicated that HFDRES EDL fatigued quicker between WL 5 and 14

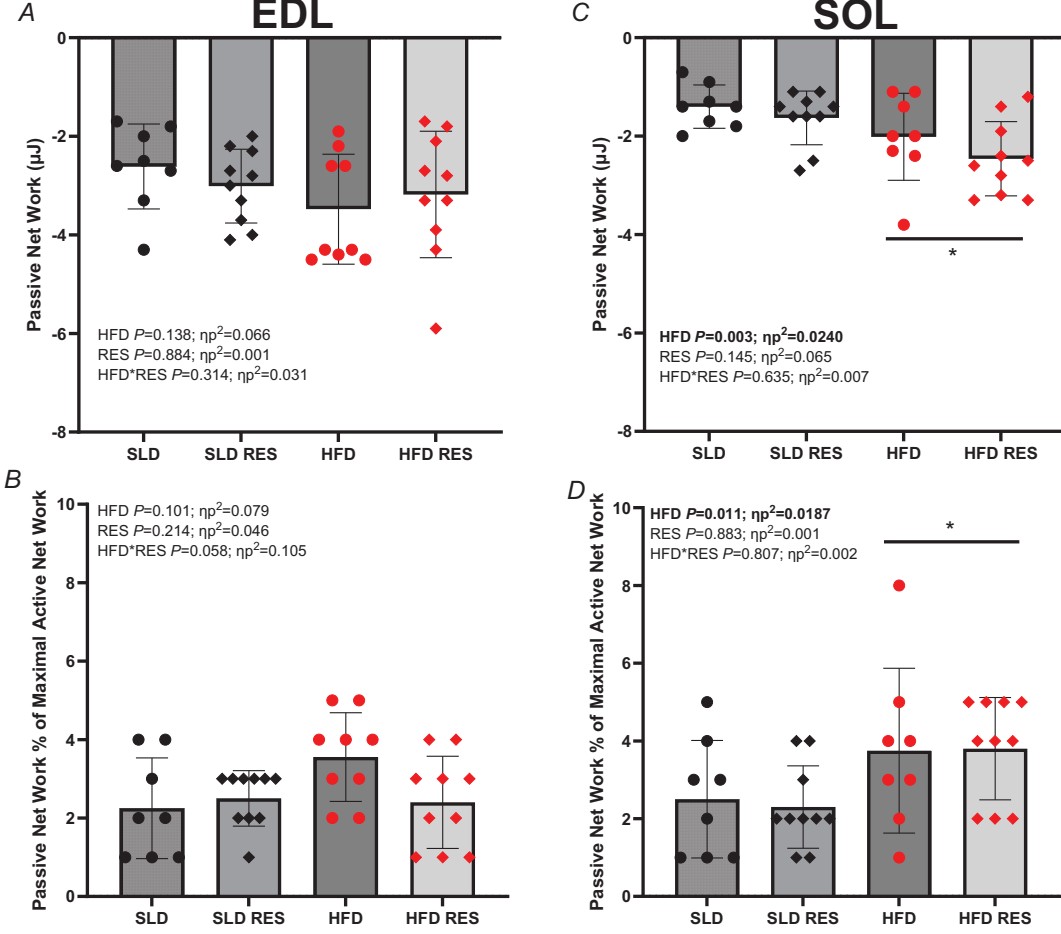

**Figure 4. The effect of 12 weeks high-fat diet and resveratrol on isolated mouse EDL and SOL passive net work**

The effect of 12 weeks of a high-fat diet and resveratrol on passive net work (µJ) (*A* and *C*) and passive net work as a percentage of maximal active net work (*B* and *D*) for the EDL (*A* and *B*) and SOL (*C* and *D*) at 10 Hz and 5 Hz, respectively. EDL, *n* = 10 for SLD RES and HFDRES, *n* = 8 for SLD and *n* = 9 for HFD; SOL *n* = 10 for SLD RES and HFDRES and *n* = 8 for SLD and HFD. Data are presented as individual data points with line and error bards representing the mean ± SD; *significant difference between HFD groups and SLD groups at *P* < 0.011. [Colour figure can be viewed at wileyonlinelibrary.com]

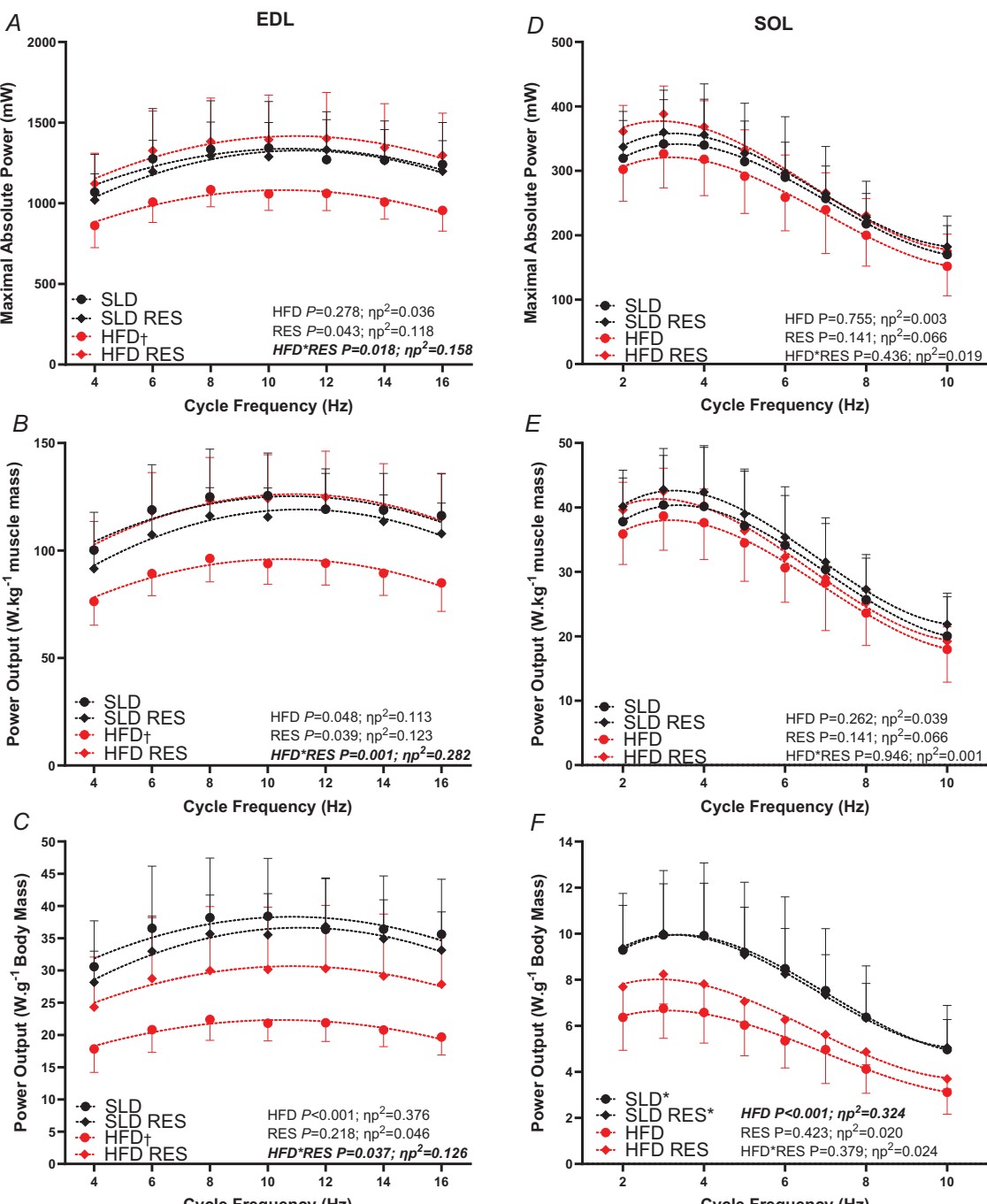

**Figure 5. The effect of 12 weeks high-fat diet and resveratrol on isolated mouse EDL and SOL work loop power**

The effect of 12 weeks of a high-fat diet and resveratrol on the work loop power output-cycle frequency relationship for absolute power output (mW), power output normalised to muscle mass (W kg$^{-1}$ muscle mass) and power output normalised to body mass (W g$^{-1}$ body mass) of isolated mouse EDL (*A–C*, respectively) and SOL (*D–F*, respectively). EDL, *n* = 10 for SLD RES and HFDRES, *n* = 8 for SLD and *n* = 9 for HFD; SOL, *n* = 10 for SLD RES and HFDRES and *n* = 8 for SLD and HFD. Data are presented as the mean ± SD; †significant difference between HFD only compared to all other groups at *P* < 0.037; *significant difference between HFD groups and SLD groups at *P* = 0.008; error bars shown in one direction only for clarity. [Colour figure can be viewed at wileyonlinelibrary.com]

compared to all other groups ($P < 0.001$) (Fig. 7*A*). A significant RES × HFD interaction was observed for cumulative work after 50 consecutive WLs ($P = 0.049$, $\eta p^2 = 0.116$) (Fig. 7*B*). Bonferroni multiple comparisons indicated that EDL cumulative work in HFD only group was lower compared to all other groups (HFD *vs.* SLD: $P = 0.009$, $d = 1.16$; HFD *vs.* HFDRES: $P = 0.037$, $d = 1.19$), but other groups did not differ (SLD *vs.* SLDRES: $P = 0.477$, $d = 0.31$; SLDRES *vs.* HFDRES: $P = 0.986$, $d = 0.01$). ANOVA SPM[*F*] also indicated significant effect of experimental group on cumulative work production for the EDL ($P < 0.001$) (Fig. 7*B*). Further exploration with the SPM[*t*] *t* test indicated that, for EDL cumulative work, the HFD group produced less cumulative work compared to HFDRES and SLD groups between WL 1 and 28 and WL 1 and 31, respectively ($P < 0.001$) (Fig. 7*B*). For recovery of maximal PO, there was a significant main effect of time ($P < 0.001$, $\eta p^2 = 0.630$) (Fig. 7*C*) indicating significant recovery of maximal PO every 10 min. There were no significant effects of HFD ($P = 0.144$, $\eta p2 = 0.079$), RES ($P = 0.181$, $\eta p2 = 0.057$) or HFD × RES ($P = 0.829$, $\eta p^2 = 0.002$) time × HFD ($P = 0.657$, $\eta p^2 = 0.007$), time × RES ($P = 0.773$, $\eta p^2 = 0.003$) or time × HFD × RES ($P = 0.219$, $\eta p^2 = 0.049$) (Fig. 6*C* and *E*) interactions.

For the SOL, there were no significant effects of HFD ($P = 0.368$, $\eta p^2 = 0.025$), RES ($P = 0.850$, $\eta p^2 = 0.001$) or HFD × RES interaction ($P = 0.688$, $\eta p^2 = 0.005$) (Fig. 7*D*) on time to 50% fatigue. ANOVA SPM[*F*] for percentage decline of PO relative to maximum indicated no effect of experimental group ($P > 0.999$) (Fig. 7*D*). HFD groups produced significantly lower cumulative work after 50 consecutive WLs compared to SLD groups

($P = 0.014$, $\eta p2 = 0.175$) (Fig. 7*E*). There was no significant main effect of RES ($P = 0.315$, $\eta p2 = 0.031$) or a RES × HFD interaction ($P = 0.963$, $\eta p2 < 0.001$). ANOVA SPM[*F*] indicated no effect of experimental group on cumulative work production ($P > 0.999$) (Fig. 7*E*). For recovery of maximal PO, there were no significant effects of time ($P = 0.088$, $\eta p^2 = 0.079$), HFD ($P = 0.166$, $\eta p^2 = 0.059$), RES ($P = 0.915$, $\eta p^2 < 0.001$) or HFD × RES ($P = 0.966$, $\eta p^2 < 0.001$) time × HFD ($P = 0.886$, $\eta p^2 = 0.003$), time × RES ($P = 0.874$, $\eta p^2 = 0.003$) or time × HFD × RES ($P = 0.889$, $\eta p^2 = 0.002$) (Fig. 7*F*) interactions.

Figures 8 illustrates typical WL shapes at standardised times during the fatigue protocol for mouse SOL (WL 2, 10, 20, 30 and 40) and EDL (WL 2, 10, 18 and 26) in each respective group. WL shapes were plotted as force against strain (%$L_0$) for the individual force and length data points for each WL cycle. The area within the WL represents the net work done during the length change cycle. The area within the typical EDL WL traces (Fig. 7*A–D*) appears initially smaller in the HFD only groups (Fig. 7*C*) compared to all other groups (Fig. 7*A*, *B* and *D*), but the shape of the loop is consistent across groups. The reduction of the area within the loop, and subsequent change in WL shape exhibited during the fatigue protocol appears uniform irrespective of diet or treatment. However, typical SOL WL shapes (Fig. 7*E–H*) demonstrate that, at the later stages of the fatigue protocol (WL 30 and onward), there is greater force production during muscle re-lengthening in HFD groups (Fig. 7*G* and *H*) compared to SLD groups (Fig. 7*E* and *F*). As a result, from WL 30, there is a marked reduction in the size of WL shape relative to the initial WL in HFD groups.

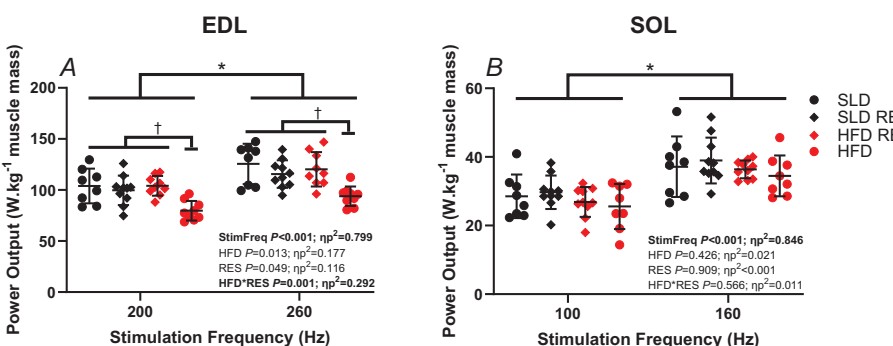

**Figure 6. The effect of stimulation frequency, and 12 weeks high-fat diet and resveratrol on the maximal work loop power output of mouse EDL and SOL**
The effect of stimulation frequency and 12 weeks of a high-fat diet and resveratrol on maximal work loop power output normalised to muscle mass (W kg$^{-1}$ muscle mass) of isolated mouse EDL (*A*) and SOL (*B*) EDL, $n = 10$ for SLD RES, $n = 8$ for SLD and $n = 9$ for HFD and HFDRES; SOL, $n = 10$ for SLD RES and HFDRES and $n = 8$ for SLD and HFD. Data are presented as individual data points and the mean ± SD; *significant difference between stimulation frequencies $P < 0.001$; †significant difference between HFD only compared to all other groups at $P < 0.002$. [Colour figure can be viewed at wileyonlinelibrary.com]

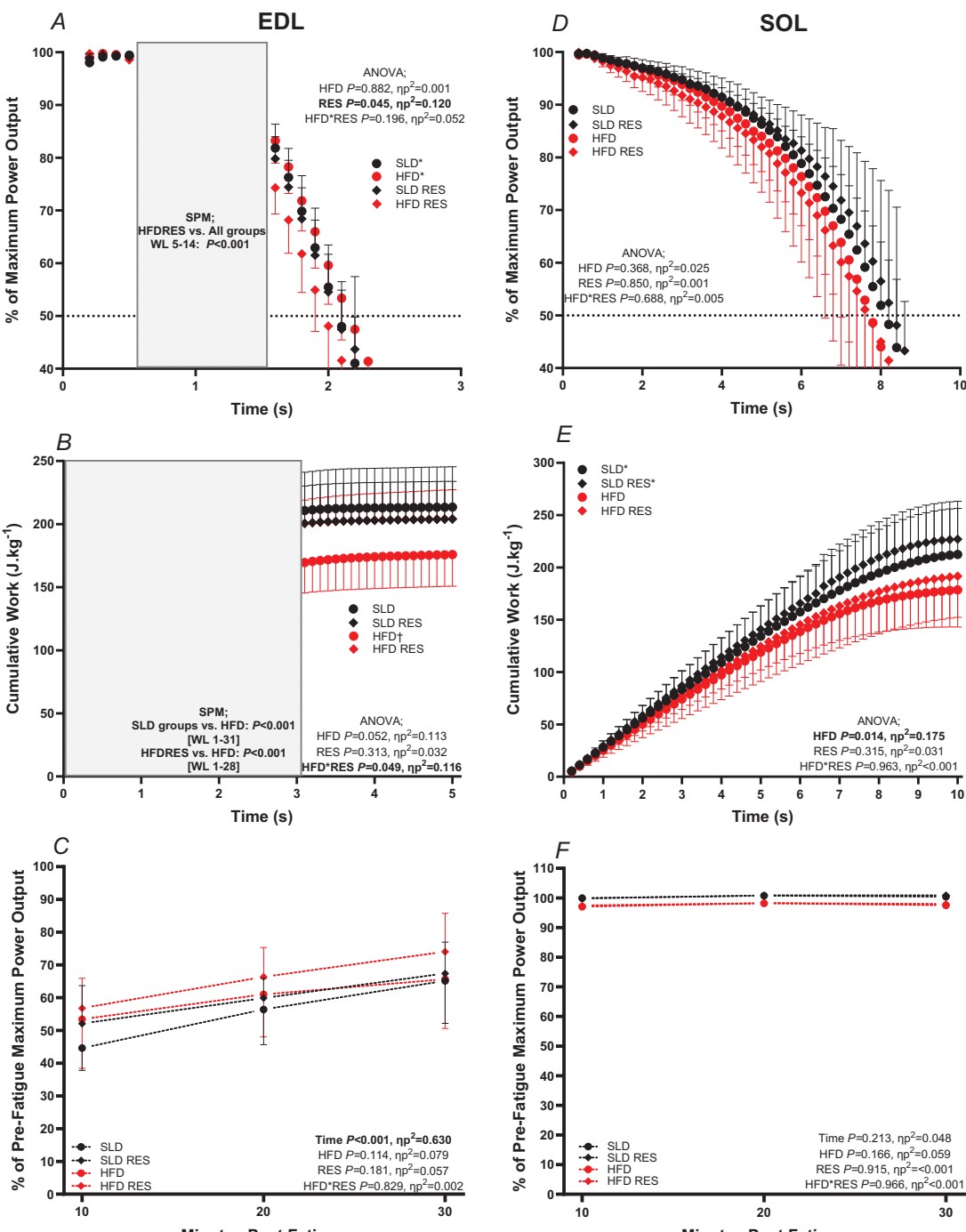

**Figure 7. The effect of 12 weeks high-fat diet and resveratrol on fatigue resistance and recovery of isolated mouse EDL and SOL**

The effect of 12 weeks of a high-fat diet and resveratrol on net muscle power output relative to maximum power output during fatigue (*A* and *D*) on cumulative work normalised to muscle mass (J kg$^{-1}$ muscle mass) produced during 50 consecutive work-loop cycles (*B* and *E*) and on the recovery of maximal work loop power output over 30 min post fatigue (*C* and *F*) of isolated mouse EDL (*A*, *B* and *C*) and SOL (*D*, *E* and *F*); shaded area of (*A*) and (*B*) indicates significant difference between HFDRES and all other groups (*A*), as well as HFD and all other groups (*B*), determined via two-sample SPM[*t*] statistical analysis. EDL, *n* = 10 for SLDRES, *n* = 8 for SLD and *n* = 9 for HFD and HFDRES; SOL, *n* = 10 for SLDRES and HFDRES and *n* = 8 for SLD and HFD. Data are presented as the mean ± SD; *significant difference between standard and RES EDL (*A*) at *P* = 0.045 and SLD and HFD SOL (*E*) at *P* = 0.014 determined via two-way ANOVA; †significant difference between HFD only compared to all other groups at *P* < 0.037. Error bars presented in one direction only for clarity. [Colour figure can be viewed at wileyonlinelibrary.com]

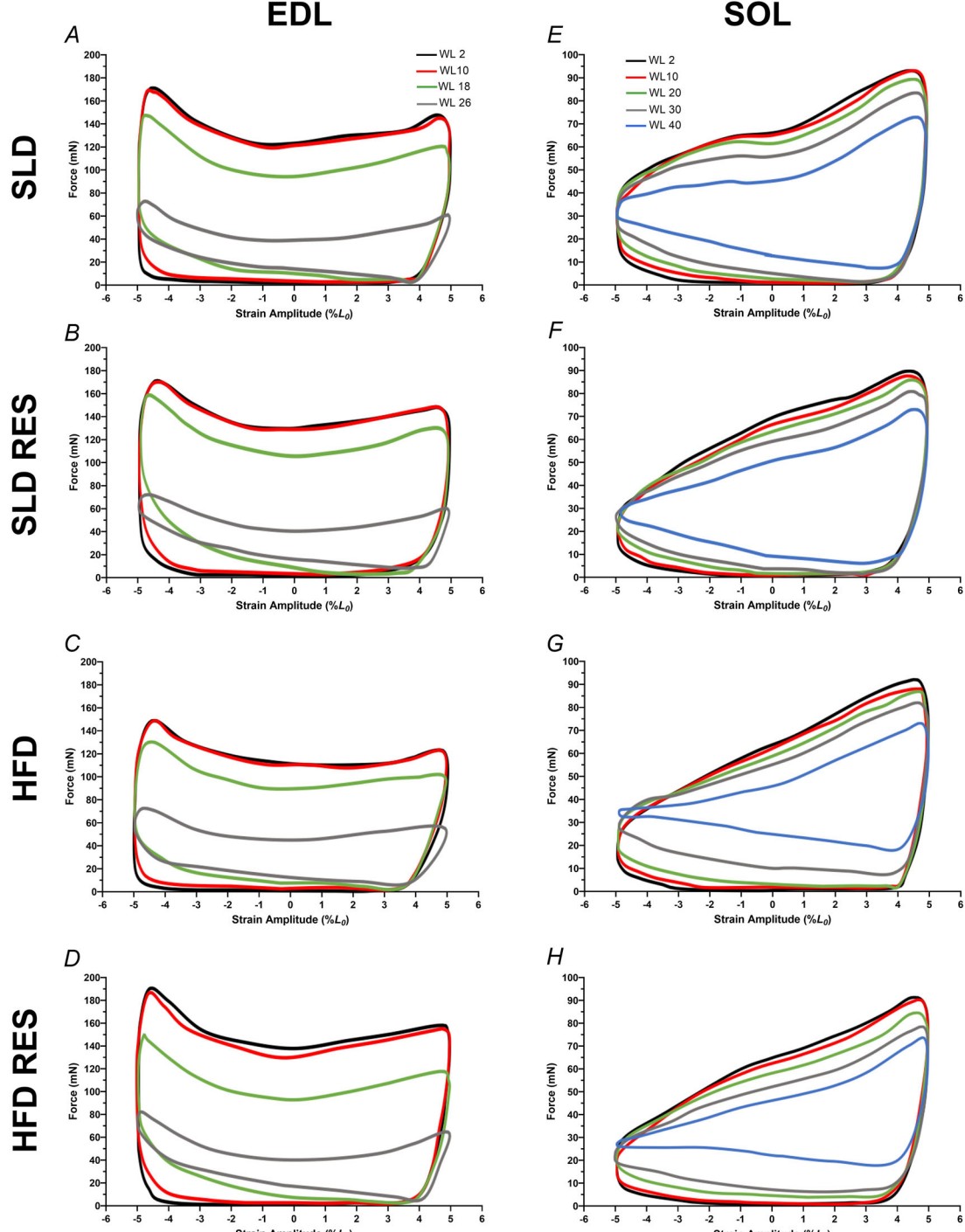

**Figure 8. Effects of 12 weeks high-fat diet and resveratrol on work loop shapes during assessment of fatigue**

Effects of 12 weeks of a high-fat diet and resveratrol (*A* and *E*, SLD; *B* and *F*, SLDRES; *C* and *G*, HFD; *D* and *H*, HFDRES) on average work loop shapes during assessment of fatigue resistance at 10 Hz and 5 Hz cycle frequency for isolated mouse EDL (*A* and *D*) and SOL (*E* and *H*), respectively. Values plotted as force against strain (%$L_0$). Work loops, 2, 10, 18 and 26 and 2, 10, 20, 30 and 40 of the fatigue protocols are shown for EDL and SOL respectively. Work loops proceed in an anti-clockwise direction, with the work loop starting at $L_0$. [Colour figure can be viewed at wileyonlinelibrary.com]

## Sirtuin 1 expression

For EDL and SOL SIRT1:GAPDH protein expression, there were no significant main effects of HFD ($P > 0.322$, $\eta p^2 < 0.049$), RES ($P > 0.815$, $\eta p^2 < 0.003$) or HFD × RES interaction ($P > 0.165$, $\eta p^2 < 0.094$) (Fig. 9*A* and *B*).

## Discussion

The present study is the first to directly examine the effects of RES supplementation on the contractile mechanics of isolated fast and slow twitch SkM from SLD and HFD mice. This data indicates that HFD consumption has limited effect on the isometric properties of isolated SkM but leads to reductions in normalised WL PO and cumulative work during fatiguing contractions of both the SOL and EDL. Furthermore, a HFD evokes a reduction in absolute PO and muscle quality (PO normalised to muscle mass) of the EDL. When consumed with a HFD, RES attenuated the HFD-induced reductions in EDL PO, without alterations in relaxation kinetics or SIRT1 expression. However, RES had little effect on contractile performance in SLD SkM. These data infer that RES may be an effective nutritional strategy to abate HFD-induced reductions in fast twitch SkM function.

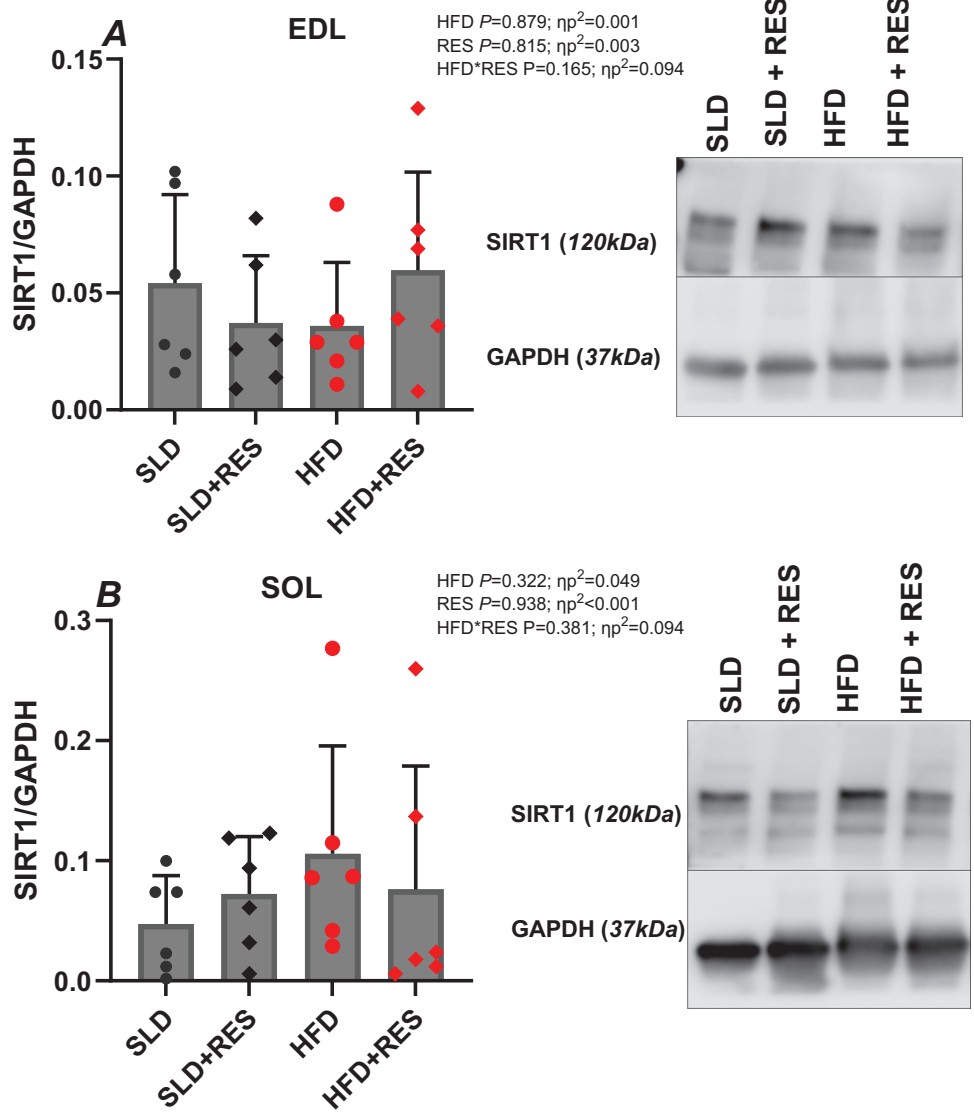

**Figure 9. The effect of 12 weeks high-fat diet and resveratrol on the ratio of SIRT1/GAPDH of isolated mouse EDL (A) and SOL (B)**

The effect of 12 weeks of a high-fat diet and resveratrol on the ratio of SIRT1/GAPDH of isolated mouse EDL (*A*) and SOL (*B*). For each muscle and treatment group, *n* = 6. Data are presented as individual data points and the mean ± SD. [Colour figure can be viewed at wileyonlinelibrary.com]

## HFD effects on isolated SkM function

It should be noted that the effects of HFD consumption on contractile performance of isolated SkM and the associated mechanisms of action have been shown to be influenced by study design heterogeneity (e.g. sex, age, strain of rodent, muscle phenotype, contractile mode, feeding duration, dietary composition and experimental test temperature), often making direct comparisons between studies difficult (Tallis et al., 2018, 2022).

The present data add further evidence indicating that HFD consumption adversely effects fast twitch EDL muscle mechanics, such that relaxation kinetics (Bott et al., 2017; Shelley et al., 2023), absolute power production (Shelley et al., 2023; Tallis et al., 2024) and muscle quality (Hurst et al., 2019; Shelley et al., 2023; Tallis et al., 2017, 2024) are all diminished. Although the present data suggest that HFD has little effect on EDL isometric force and stress, this is not always the case in studies utilising feeding durations $\geq$12 weeks (Eshima et al., 2017, 2020; Shelley et al., 2023; Tallis et al., 2017). However, the data infer a similar trend indicating a HFD-induced reduction in EDL tetanic stress (SLD: 332.7 $\pm$ 63.6; HFD: 283.4 $\pm$ 52.0) at a moderate effect size ($d = 0.85$). Importantly, this study is the first to indicate that HFD does not affect passive work of the EDL, suggesting that HFD-induced changes in EDL PO are primarily a result of a reduction in the ability of the muscle to produce work during shortening (i.e. active work). It has been speculated that the adverse effects of HFD consumption are more substantial in fast-twitch SkM, partly as the result of an unfavourable metabolic profile for oxidising lipids (Tallis et al., 2017).

Comparatively, HFD effects on SOL contractile performance are more ambiguous. In accordance with previous work, HFD did not affect activation and relaxation kinetics (Bott et al., 2017; Ciapaite et al., 2015; Eshima et al., 2020; Shelley et al., 2023; Tallis et al., 2022), absolute force and power or PO–CF curve (Eshima et al., 2020; Hurst et al., 2019; Shelley et al., 2023), isometric stress (Eshima et al., 2020; Hurst et al., 2019; Shelley et al., 2023; Tallis et al., 2017, 2022, 2024) or muscle quality (Shelley et al., 2023; Tallis et al., 2017, 2024). However, these results add new insight indicating that HFD increased the negative work of SOL during passive WL cycles. Nevertheless, this increased passive stiffness was not substantial enough to culminate in reduced maximal WL PO in electrically stimulated WLs. It important to note that the response of SOL to HFD feeding is influenced by feeding duration (Hurst et al., 2019). For example, longer HFD feeding durations ($\geq$16 weeks) can result in an increase in absolute SOL tetanic force, as suggested to occur through a training adaptation from prolonged stabilising of a larger mass given the role of SOL in postural control (Tallis et al., 2017, 2022, 2024). Antithetically, shorter feeding durations appear to diminish SOL contractile performance. In previous work, SOL muscle quality was reduced after 8 weeks of a HFD, but not 12 weeks (Hurst et al., 2019), and SOL tetanic force normalised to muscle mass reduced after 5 weeks of high-fat palm oil but not high-fat lard (Ciapaite et al., 2015). As such, future work is needed to provide mechanistic insight into the duration/diet-specific responses to HFD feeding that underpin altered SOL contractile performance.

Although the magnitude and extent of adverse effects of HFD consumption on acute power production are muscle specific, HFD-induced changes in fatigue resistance initially appear consistent between muscles. The present data indicate that the rate of fatigue in both SOL and EDL is unaffected by HFD, which has been reported previously (Hurst et al., 2019; Shelley et al., 2023; Tallis et al., 2017). However, the present study extends the current understanding. Despite no change in rate of fatigue, cumulative work production is diminished in HFD SOL and EDL. Cumulative work is important when considering *in vivo* function because the amount of work, in addition to the rate of fatigue, is influential to activities of daily living that require repetitive muscle actions. *In vivo*, the adverse effects of HFD on cumulative work are probably compounded when the isolated muscle that already has diminished ability to produce work over consecutive actions is required to move a larger body mass.

EDL WL shapes (Fig. 8*A–D*) infer that the HFD-induced reduction in cumulative work is related to the reduction in work during shortening. HFD and SLD EDL fatigue at a similar rate and produce WLs that are comparable in shape but differ in size, indicating reduced net work. The initial reduction in net work in HFD EDL is compounded throughout fatigue, resulting in a reduction of total cumulative work. Given acute SOL PO was comparable between SLD and HFD groups, similar effects do not explain impaired cumulative work production in HFD SOL. Visual analysis of WL shapes indicate that, at the later stages of the fatigue protocol, HFD SOL has a larger negative work component during re-lengthening (Fig. 8*E–H*), as indicated by force production during the re-lengthening phase. An increased negative work component would contribute to a reduction in total net work per cycle (net work per cycle = total work – negative work). This increased negative work is indicative of prolonged relaxation time, which is a suggested consequence of HFD-induced impairment of sarco(endo)plasmic reticulum $Ca^{2+}$-ATPase (SERCA) function (Funai et al., 2013). SERCA is responsible for returning $Ca^{2+}$ back into the sarcoplasmic reticulum causing muscle relaxation; HFD-impaired SERCA function in the SOL could result in the quantity of $Ca^{2+}$ in the cytoplasm to compound and exacerbate as the fatigue protocol progresses, corresponding with

continually increasing negative work component with each subsequent contraction. Changes in the SkM fatigue response are often complex and cannot be attributed to single factors. Therefore, it is important to recognise that factors such as a HFD-induced reduction in the efficacy of actin–myosin cross-bridge cycling (Ciapaite et al., 2015; Schilder et al., 2011) could also evoke HFD-induced alterations in fatigue mechanics of both the SOL and EDL, independent of changes in acute force and power production.

## Resveratrol effects on maximal force, power and fatigue resistance

Supplementation of RES consumed with SLD had little effect on acute force and power production of isolated SkM. However, when consumed with a HFD, RES had contractile mode and muscle-specific effects that were more apparent in fast twitch EDL muscle. In EDL, RES alleviated the adverse effects of a HFD on absolute and normalised PO and muscle quality (power normalised to muscle mass). The attenuation of HFD-induced reductions in contractile performance through consumption of RES are probably more pronounced in isolated EDL because HFD effects are more substantial in fast twitch muscle. Mechanistically, RES could attenuate the adverse effects of HFD on EDL PO through altering fibre type composition (fast to slow) (Huang et al., 2020; Wen et al., 2020). Previous observations indicate that HFD effects on maximal PO may partly be the result of a slow to fast shift in fibre type composition (Seebacher et al., 2017; Tallis et al., 2018). However, this would not appear to be a driving mechanism in this instance because the CFs used to elicit maximal PO, as well as the shape of the power output-cycle frequency curves (PO CF), were unaffected in any of the treatments. If substantial changes in myosin heavy chain composition occurred, one would expect a leftward (fast:slow shift) or rightward (slow:fast) shift in the whole curve and CF needed to elicit maximal PO, as demonstrated in differences between PO CF curves of the fast twitch EDL and slow twitch SOL (Fig. 5). Furthermore, previous work utilising isolated SkM from female CD-1 mice of similar ages to those used in the present study reported no changes in myosin heavy chain composition between SLD and HFD EDL (Messa et al., 2020; Tallis et al., 2017). Therefore, based on these data indicating reduced accumulation of adipose in HFD when supplemented with RES (Table 1), it is more probable that one of the key factors reducing the HFD-induced reduction in EDL PO is through RES limiting the magnitude of adipose (Cho et al., 2012; Kim et al., 2011; Lagouge et al., 2006; Shabani et al., 2020) and intramuscular lipid accumulation (Shabani et al.,

2020). Given that central adipose tissue was still greater in HFDRES compared to SLD groups, lipid metabolism alone is probably not the only mechanism attenuating contractile performance.

The attenuation of HFD-induced effects on SkM performance evoked through RES supplementation, was in part hypothesised as a result of RES activating SIRT1 (Pfluger et al., 2008; Price et al., 2012). When SIRT1 and closely associated molecular pathways (e.g. AMPK and nuclear factor kappa B) are inhibited through obesity, excessive fat deposition can occur through impaired lipid metabolism and increased chronic low-grade inflammation and oxidative stress (Fu et al., 2013; Lyons & Roche, 2018; Pardo & Boriek, 2020; Steinberg et al., 2006; Tallis et al., 2018), comprising factors that mediate obesity-induced SkM dysfunction. Previous observations suggest that RES activates SIRT1 in the biological tissue of individuals with obesity and HFD rodents (Pfluger et al., 2008; Price et al., 2012; Timmers et al., 2011), which in turn protects against HFD-induced metabolic damage through preserved lipid and energy metabolism and reduced inflammation (Pfluger et al., 2008). However, in the present study, SIRT1 expression was unchanged irrespective of treatment (Fig. 8). This may indicate that SIRT1 is not a contributing factor to persevered HFD EDL contractile performance and that RES-induced activation of alternative pathways contributed to persevered SkM performance. RES may in fact offset the HFD-induced reduction in AMPK activity (Shabani et al., 2020), a whole body regulatory system that controls several mechanisms contributing to SkM health, including lipid metabolism and protein synthesis (Wang et al., 2018). Consequently, enhanced AMPK activity reported in HFD muscle when supplemented with RES (Shabani et al., 2020), at an identical dose as that in the present study (4 g kg$^{-1}$ diet), could be a driving mechanism for maintained EDL PO. It should be noted that RES-induced activation of AMPK has previously shown to be SIRT1-dependent (Price et al., 2012). Therefore, albeit speculative, it could be that there is a transient change in expression of SIRT1, where the SIRT1 protein is expressed during the initial phases of treatment leading to attenuation of HFD effects, which may not be apparent at the end of the treatment. However, to date, there is no evidence directly linking the effects of RES on contractile performance with the mechanisms mediating SkM function. As such, why RES abates the adverse effects of a HFD on EDL PO remains speculative and warrants further investigation.

In EDL, RES treated muscles fatigued quicker compared to controls. However, it is important to note that TT50% fatigue between SLD groups (SLD: 2.15 ± 0.12 s; SLDRES: 2.11 ± 0.16 s) appears comparable. The main effect observed should be interpreted with caution because it could have been influenced by HFDRES EDL, which fatigued at a quicker rate than all other groups between

WL 5 and 15, as identified through statistical parametric mapping; the magnitude of difference was not great enough to achieve an interaction in the two-way ANOVA. Irrespective of potential RES-induced changes in rate of fatigue, this did not adversely impact cumulative work production. Indeed, HFDRES alleviated the reduction in cumulative work established in HFD EDL. The preservation of acute power in HFDRES EDL appears to translate into greater cumulative work during fatiguing contractions because greater work is produced per WL cycle. Increased capacity to produce work during fatiguing contractions could promote increased exercise capacity and the ability to complete activities of daily living that require bouts of consecutive contractions, in turn increasing the capacity the expend energy promoting a calorie deficit.

In SOL, RES did not attenuate any HFD effects on PO normalised to body mass or cumulative work production during fatigue, but did evoke an increase in absolute SOL tetanus force in the HFDRES group compared to HFD only. The effect may be explained by maintenance of processes involved in myogenesis as indicated by previous work that demonstrates promotion of SkM myogenesis and hypertrophy *in vitro* (Montesano et al., 2013), resulting in more effective adaptation to the elevated load (e.g. greater body mass in HFD groups) placed on postural muscles. The non-significant but average increase in CSA ($\sim$9.1%; $d = 0.73$) in HFDRES SOL compared to HFD only supports the notion of effective adaptation to larger body mass and could partly explain the increased absolute force, also providing a plausible explanation for why, despite an increase in absolute force, no effect on isometric stress was identified. However, SOL stress was $\sim$14.7% greater in HFDRES compared to HFD with a moderate effect size observed ($d = 0.88$).

### Limitations and future directions

Although the present study provides valuable insights into the potential of RES for reducing the impact of HFD consumption on SkM function, it is not without limitations. One limitation of the present study is providing supplementation of RES through dietary enrichment in grouped cages, as opposed to a method where accurate dosage per animal can be monitored, such as oral gavage or individual housing. Dosing through direct methods ensures accurate measures of the dose of supplement being provided to each individual mouse. However, both individual housing and oral gavage can induce significant stress upon the animal (Bonnichsen et al., 2005; Brown et al., 2000; Manouze et al., 2019), which in this instance can be mitigated by providing the RES as part of their diet, in cages containing multiple mice. Although it was not possible to determine the exact

quantity of RES consumed, the approximation was based on previous research that reported that RES enriched into a diet at 4 g kg$^{-1}$, as used in the present study, equates to an approximate daily dose of 400 mg kg$^{-1}$ body mass (Lagouge et al., 2006).

Although the WL model provides a close approximation of *in vivo* dynamic muscle activity, the sinusoidal length change waveforms employed are a simplification of the muscle-specific length change waveforms used *in vivo* (Dickinson et al., 2000). Furthermore, fibre stimulation and length change waveform are probably manipulated during fatiguing contractions *in vivo* to optimise muscle performance (Wakeling & Rozitis, 2005). The WL model does not reflect the more complex mechanical characteristics of fatiguing muscle actions occurring *in vivo*.

Although the present study provides an insight into the potential therapeutic use of RES for alleviating the detrimental effects of a HFD on contractile performance of SkM, there are still several avenues for future research. First, the present study provides an insight into SIRT1 expression of the SOL and EDL, but additional exploration of the underpinning mechanisms behind RES mediation of HFD-induced reductions in SkM function (e.g. histological analysis of tissue to quantify lipid infiltration, fibre type composition and size, and capillary distribution and size) is needed. Additionally, the present study was conducted in female mice to allow for direct comparisons from previous work examining the effects of HFD on isolated SkM contractility using a similar methodological approach. However, future work is needed to directly determine the sex-specific effects of HFD consumption on isolated SkM contractility and ascertain whether RES can be effective in alleviating HFD-induced declines in isolated SkM function of male mice. Although the dose used in the present study attenuated HFD-induced reductions in SkM function, it is yet to be determined whether changes in muscle contractility are dose-dependent. Recent research also identified that HFD-induced reductions in muscle performance are contractile mode and muscle specific, that is muscle-specific changes in concentric muscle quality are not always associated with changes in eccentric muscle quality and *vice versa* (Shelley et al., 2021; Tallis et al., 2024). Adequate eccentric force is important for maintaining balance, deceleration and absorbing impact, which occurs during activities such as stair descents, descending into a seated position and during activities that require dynamic balance (Delbaere et al., 2003; Nishikawa, Lindstedt, et al., 2018). As such, future work should consider the therapeutic effects of RES on a HFD-induced reduction in eccentric performance of locomotor muscles. Finally, because the present data indicate that RES may reduce the impact of HFD consumption on contractile performance of isolated SkM, future work should explore the therapeutic effects

of RES on SkM health in other disease models that evoke significant reductions in muscle quality (e.g. cancer and ageing).

## Conclusions

In summary, when consumed with a HFD, RES reduced the accumulation of adipose tissue and ameliorated the reduction in absolute and normalised PO, as well as cumulative work during fatiguing contractions, of fast-twitch EDL. When RES was consumed with SLD, RES had limited effects on contractile function, irrespective of the mode of contractility or muscle fibre type composition. These pre-clinical data suggest that RES may be a low-cost and appealing nutritional strategy to offset obesity-induced SkM pathology, as well as important for improving physical function, attenuating a negative obesity cycle and ultimately promoting whole body health.

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

## Additional information

### Data availability statement

All data associated with this study are presented in the article or are available via FigShare (10.6084/m9.figshare.25869097)

### Competing interests

The authors declare that they have no competing interests.

### Author contributions

S.S., J.T. and R.S.J. conceived and designed the study. S.S. and M.T. performed data collection. S.S. and J.T. analysed the data. S.S. prepared figures. J.T., R.S.J., S.E. and E.E. supervised the project. S.S. and J.T. drafted manuscript. All authors edited and revised the manuscript. All authors have approved the

final version of the manuscript submitted for publication and agreed to be accountable for all aspects of the work. All persons designated as authors qualify for authorship, and all those who qualify for authorship are listed.

## Funding

No external funding was received for this work.

## Acknowledgements

We thank the members of The University of Warwick's Biological Services Unit for their support and care of the animals throughout the duration of this study.

## Keywords

fatigue, force, muscle function, nutraceuticals, obesity, power output, work loop

## Supporting information

Additional supporting information can be found online in the Supporting Information section at the end of the HTML view of the article. Supporting information files available:

**Peer Review History**

