## [Peer Review History · The Journal of Physiology]

Adverse Effects of High-Fat Diet Consumption on Contractile Mechanics of Isolated Mouse Skeletal Muscle are Reduced When Supplemented with Resveratrol

Sharn Shelley, Rob S James, Steven J Eustace, Mark C Turner, Ryan Brett, Emma Eyre, and Jason Tallis
DOI: 10.1113/JP287056

Corresponding author(s): Jason Tallis (ab0289@coventry.ac.uk)

Review Timeline:

Submission Date:	04-Jun-2024
Editorial Decision:	18-Jul-2024
Revision Received:	03-Feb-2025
Editorial Decision:	06-Mar-2025
Revision Received:	24-Mar-2025
Accepted:	27-Mar-2025

Senior Editor: Karyn Hamilton

Reviewing Editor: Nathan Winn

Transaction Report:

Dear Dr Shelley,

Re: JP-RP-2024-287056 "Adverse Effects of High-Fat Diet Consumption on Contractile Mechanics of Isolated Mouse Skeletal Muscle are Reduced When Supplemented with Resveratrol" by Sharn Shelley, Rob S James, Steven J Eustace, Mark C Turner, Ryan Brett, Emma Eyre, and Jason Tallis

Thank you for submitting your manuscript to The Journal of Physiology. It has been assessed by a Reviewing Editor and by 2 expert referee and we are pleased to tell you that it is potentially acceptable for publication following satisfactory major revision.

LANGUAGE EDITING AND SUPPORT FOR PUBLICATION: If you would like help with English language editing, or other article preparation support, Wiley Editing Services offers expert help, including English Language Editing, as well as translation, manuscript formatting, and figure formatting at www.wileyauthors.com/eoo/preparation. You can also find resources for Preparing Your Article for general guidance about writing and preparing your manuscript at www.wileyauthors.com/eoo/prepresources.

REVISION CHECKLIST:

We look forward to receiving your revised submission.

Yours sincerely,

Karyn Hamilton
Senior Editor
The Journal of Physiology

REQUIRED ITEMS

- Author photo and profile. First or joint first authors are asked to provide a short biography (no more than 100 words for one author or 150 words in total for joint first authors) and a portrait photograph. These should be uploaded and clearly labelled together in a Word document with the revised version of the manuscript. See Information for Authors for further details.

- Please upload separate high-quality figure files via the submission form.

- You must upload original, uncropped western blot/gel images (including controls) if they are not included in the manuscript. This is to confirm that no inappropriate, unethical or misleading image manipulation has occurred. These should be uploaded as 'Supporting information for review process only'. Please label/highlight the original gels so that we can clearly see which sections/lanes have been used in the manuscript figures. For more information, see: <https://physoc.onlinelibrary.wiley.com/hub/journal-policies#imagmanip>.

- Please ensure that the Article File you upload is a Word file.

- Papers must comply with the Statistics Policy: https://jp.msubmit.net/cgi-bin/main.plex?form_type=display_requirements#statistics.

In summary:

- If $n \leq 30$, all data points must be plotted in the figure in a way that reveals their range and distribution. A bar graph with data points overlaid, a box and whisker plot or a violin plot (preferably with data points included) are acceptable formats.

- If $n > 30$, then the entire raw dataset must be made available either as supporting information, or hosted on a not-for-profit repository, e.g. FigShare, with access details provided in the manuscript.

- 'n' clearly defined (e.g. x cells from y slices in z animals) in the Methods. Authors should be mindful of pseudoreplication.

- All relevant 'n' values must be clearly stated in the main text, figures and tables.

- The most appropriate summary statistic (e.g. mean or median and standard deviation) must be used. Standard Error of the Mean (SEM) alone is not permitted.

- Exact p values must be stated. Authors must not use 'greater than' or 'less than'. Exact p values must be stated to three significant figures even when 'no statistical significance' is claimed.

- Please include an Abstract Figure file, as well as the Figure Legend text within the main article file. The Abstract Figure is a piece of artwork designed to give readers an immediate understanding of the research and should summarise the main conclusions. If possible, the image should be easily 'readable' from left to right or top to bottom. It should show the physiological relevance of the manuscript so readers can assess the importance and content of its findings. Abstract Figures should not merely recapitulate other figures in the manuscript. Please try to keep the diagram as simple as possible and without superfluous information that may distract from the main conclusion(s). Abstract Figures must be provided by authors no later than the revised manuscript stage and should be uploaded as a separate file during online submission labelled as File Type 'Abstract Figure'. Please also ensure that you include the figure legend in the main article file. All Abstract Figures should be created using BioRender. Authors should use The Journal's premium BioRender account to export high-resolution images. Details on how to use and access the premium account are included as part of this email.

EDITOR COMMENTS

Reviewing Editor:

Dear Authors, the reviewers were, overall, enthusiastic about the execution of the experiments. However, there are some significant issues that need to be addressed. One issue raised is that the results are not appropriately discussed in the context of previous literature. Key studies on obesity-induced skeletal muscle changes and the effects of resveratrol on muscle structure/function are not referenced. Some of these references have been included by Reviewer 1. We invite you to carefully respond to reviewers comments with particular focus on integrating prior literature surrounding some of the controversies of resveratrol.

Senior Editor:

Comments for Authors to ensure the paper complies with the Statistics Policy (Required):

Thank you in advance for double checking to make sure your manuscript complies with The Journal's statistics policy. We appreciate that you've tried using precise p-values as required, but they seem to be listed with the symbols "< or >" despite the use of precise values. Some figures also still contain $P < 0.05$

Comments to the Author:

Thank you for submitting your manuscript for consideration by The Journal of Physiology. As part of the peer review process, we recruited two Referees with expertise in this field of study. As the Reviewing Editor notes, the Referees were generally quite enthusiastic about the potential for this manuscript to be influential and of interest to The Journal's readership. However, they also both raised some major concerns with would need the authors' attention before we can consider the manuscript for publication. We would like to provide you the opportunity to address every concern point-by-point, and submit a revised manuscript addressing these detailed peer critiques. Please also revisit The Journal's statistics policy to make sure your manuscript is in full compliance. Finally, please also clearly state if ethics approval was obtained (you currently just indicate that it was sought). Thank you for your interest in The Journal and we look forward to receiving your revised manuscript.

REFEREE COMMENTS

Referee #1:

The authors address a relevant physiological question and provide new evidence that resveratrol supplementation diminishes the negative effects of a high-fat diet on muscle contractile function in mice. However, I believe manuscript requires some major and minor corrections.

Major comments:

1) The manuscript only references studies that support resveratrol as a potential treatment for obesity-induced skeletal muscle impairments (lines 99-108). However, this topic remains controversial. While animal studies suggest that resveratrol may improve skeletal muscle structure, function, and performance, clinical trials have yet to provide definitive evidence (Poulsen MM et al., Diabetes 2013; Toniolo L et al., Nutrients 2023). This contradicts what is mentioned in the manuscript (lines 107 and 108).

2) Please indicate in the introduction whether previous studies include or lack in vivo, in situ, or in vitro functional experiments, along with the advantages and disadvantages of each approach. This will help clarify the knowledge gap the study aims to address. E.g. previous experiments show that resveratrol improves muscle strength (i.e., grip strength) in rats with high-fat diet-induced obesity (Huang Y et al., Aging (Albany NY) 2019). This is an important finding that is not mentioned in the manuscript and although muscle strength and muscle contractile function are distinct concepts, they are related, and including this will help clarify the novelty of the paper. Some clear rationale is also needed in the abstract (lines 30-34).

3) Potential mechanisms for resveratrol benefits in skeletal muscle are mentioned in the manuscript (lines 102-107, 112-114, 128-130), but these are poorly described. Please provide a more detailed description of these mechanisms, particularly those related to Sirtuin1. This will help readers to understand the rationale for measuring Sirtuin1. It is also important to show the controversies regarding Sirtuin1 activation by resveratrol. Resveratrol is not a direct Sirt1 activator, and attributing its effects in high fat-fed mice to Sirt1 without genetic validation has been problematic (Brenner C, Life Metab 2022).

4) The study suggests mechanisms for resveratrol's benefits in muscle but only includes functional experiments and Sirtuin1 protein measurements. Additional molecular (e.g., gene expression and/or protein levels of Sirtuin1-related targets or other pathways) and histological (e.g., fibre typing) analyses would provide mechanistic insights and strengthen the paper.

5) Outcomes of studies on the impact of obesity on muscle contractile function have been inconsistent (Tallis J et al., J Exp Biol 2018), and this is either not mentioned or poorly discussed in the current manuscript. I suggest a more detailed discussion in the section 'High-fat Diet Effects on Isolated Skeletal Muscle Function,' considering factors such as the muscle phenotype, contractility mode, feeding period and nutrient profile of diets, as these can influence the obesity response.

Minor comments:

1) Please define obesity (e.g., increased body weight or increased adiposity), since there is evidence that increased body weight can increase muscle force and power in postural muscles (lines 65).

2) Experiments were performed in female rats only such that it remains unclear whether similar findings would be observed in males. Please explain why only female mice were included or mention this in 'limitations.'

3) Not all adverse effects of high-fat diet were reversed by resveratrol - LSHR time was longer in HFD groups compared to SLD, but similar in HFD groups with or without resveratrol. Please correct this (line 42).

4) The data mentioned in lines 180-183 should be included in the introduction to help clarify the knowledge gap between in vivo (muscle strength) and in vitro (contractile function) assessments, addressed by this study (see Major comment 2).

5) Please correct Table 1. Not all significant values are marked in bold.

6) I recommend simplifying the work loop power output results (lines 435-459). This can be organised into two paragraphs, one for the EDL data and one for the soleus data. I found lines 439-441 difficult to understand and recommend they be reformulated and simplified. The HFD*resveratrol interaction (lines 442-445) is the most relevant data here and should be mentioned first. The data in lines 435-437 seem unnecessary and can be excluded.

7) The same for the Fatigue and Recovery results. This can be organised into two paragraphs - the EDL data first followed by the soleus data.

8) I had trouble understanding why muscle fatiguability was assessed as time to 50% fatigue. Also, lines 478 and 479 state that 'for the EDL, time to 50% fatigue was significantly lower in RES treated groups compared to control groups,' but line 664 states the opposite: 'In EDL, time to fatigue was higher in RES treated muscle compared to controls.' Please correct these inconsistencies.

9) The content in lines 548-552 appears somewhat redundant and could be simplified.

Referee #2:

This ms is a report on the mechanical properties of muscles (fast-twitch extensor digitorum longus and slow-twitch soleus) isolated from mice fed a standard or high fat diet with or without Resveratrol (RES). The aim was test the hypothesis that RES would reduce the effects of the high fat diet on body fat and muscle performance, and that increased expression of SIRT1 would occur. The experiments were well designed to test these hypotheses.

The authors quite rightly point out the value of such in vitro experiments, which were ideal for discovering, for example, that the performance in the power output tests in response to diet and RES were different for the two muscle types. I'm sure this is trying to tell us something about the mechanisms of the diet and RES actions, and more discussion along these lines would have been welcome.

The points below are suggestions for improvement of the presentation.

Specific points

L88. Insert "," after Messa et al. 2020).

L103. Change "opposes" to "oppose".

L176. Insert "in vivo" after "and".

Giving that the solubility of CO₂ is strongly temperature dependent (more soluble at lower temperature), I'm curious about the pH of saline (same ionic composition and equilibrated with 5% CO₂).

L201. What was the pH during dissection of the at 1-3o?

L203. "... pH7.55 at room temperature before oxygenation". Before equilibration with 95% O₂, 5% CO₂, I would have expected CaCO₃ to have precipitated.

L216. What was the pH during the tests of contractility at 37oC?

L253. Were work-loops during movement without stimulation recorded to detect passive contributions of parallel elasticity (PE) to force? Were any measurements of series elasticity (SE) made? There seems to be no mention of parallel and/or series elasticity. Parallel elastic force would add to total force during movement; changes in the length of the series elasticity would affect the velocity of filament sliding during the work-loops (and thus the work done by the contractile component). When/if obesity and/or RES change work-loops, it would be useful to know how much, if any, of the changes are due to passive as opposed to contractile properties.

L284-285. Deteriorating performance due to anoxic core etc. and correction of the net-work. Was such a correction made? If yes, state the size of the correction.

L414 & Table 1. Clarify: Mass and csa are not larger for HFD groups compared to SLD groups? Any evidence about intramuscular fat?

L430 & Fig 3, L464 & Fig 4. Indicate the rationale for presenting the results not normalize for muscle csa or mass, respectively. I don't see the point.

L433. "*" indicates significant difference at $P < 0.05$ ". Difference between the group marked with * and which other group?

L436. Fig 3A should be Fig 4A.

L469. "*" indicates significant difference at $P < 0.05$ ". Difference between the group marked with * and which other group?

L478. Change "...lower in RES" to "...less in HFD RES treated". In Fig 6A the times look extremely similar for the SLD groups.

L542. Fig 8A & D should be Fig 8A & B.

L 556. Delete "synergistically". "...consumed with" is enough. "Synergistically consumed" does not make sense. When there is an effect of HFD and RES on muscle performance, it seems antagonistic (HFD reduces performance, and RES counteracts) rather than synergistic.

L563. Remove "to suggest".

L566. I think you mean "consistent" of "in agreement", rather than "concurrent" which means occurring at the same time.

L593. How do you know the force is "passive" rather than active (due to force produced by attached crossbridges)?

L634. See also L414 & Table 1. Do you have any evidence about intramuscular lipid accumulation?

L664. Would be clearer to say that fatigue was greater, rather than higher.

L703 & 704. Clarify what "not concurrent" means here.

END OF COMMENTS

EDITOR COMMENTS

Reviewing Editor:

Dear Authors, the reviewers were, overall, enthusiastic about the execution of the experiments. However, there are some significant issues that need to be addressed. One issue raised is that the results are not appropriately discussed in the context of previous literature. Key studies on obesity-induced skeletal muscle changes and the effects of resveratrol on muscle structure/function are not referenced. Some of these references have been included by Reviewer 1. We invite you to carefully respond to reviewers comments with particular focus on integrating prior literature surrounding some of the controversies of resveratrol.

Senior Editor:

Comments for Authors to ensure the paper complies with the Statistics Policy (Required):

Thank you in advance for double checking to make sure your manuscript complies with The Journal's statistics policy. We appreciate that you've tried using precise p-values as required, but they seem to be listed with the symbols "< or >" despite the use of precise values. Some figures also still contain $P < 0.05$.

RESPONSE: Where possible we have adhered to the statistical policy of providing the exact P value. On occasions we have provided a symbol "< or >" to denote all P values are lower or higher than that specific value on outcomes which show similar outcomes (for example, EDL power output and power output normalised to muscle mass and body mass all show a HFDxRES interaction). We have opted for this approach to keep the results as clear and concise as possible. However, we have now provided exact P values embedded into each figure. Where significant interactions occur, the multi comparisons are now given in full detail opposed to grouping of results. The only occasion where we continue to denote P values lower or higher than a specific value and have not included the full list of P values embedded into the figure is for the effect of cycle frequency. The mixed model repeated measures ANOVA for work loop power output compares power output from every cycle frequency, thus providing a P value for each comparison would result in a list of 24 comparisons, where it would be difficult to identify the trend that the statistical data demonstrates.

Comments to the Author:

Thank you for submitting your manuscript for consideration by The Journal of Physiology. As part of the peer review process, we recruited two Referees with expertise in this field of study. As the Reviewing Editor notes, the Referees were generally quite enthusiastic about the potential for this manuscript to be influential and of interest to The Journal's readership. However, they also both raised some major concerns with would need the authors' attention before we can consider the manuscript for publication. We would like to provide you the opportunity to address every concern point-by-point, and submit a revised manuscript addressing these detailed peer critiques. Please also

revisit The Journal's statistics policy to make sure your manuscript is in full compliance. Finally, please also clearly state if ethics approval was obtained (you currently just indicate that it was sought). Thank you for your interest in The Journal and we look forward to receiving your revised manuscript.

RESPONSE: The authors are pleased that both the senior and reviewing editors agree upon the value of this work and that they believe it is of interest to the journal readership. The reviewers selected have provided a thorough and insightful review which has helped us to improve the quality of our submission. We hope we have adequately amended the manuscript in line with the reviewer's comments and that it is now deemed worthy of publication. We have now amended the methods section to explicitly outline that ethical approval was granted from both the host institution and UK Home Office [please see line 194].

REFeree COMMENTS

Referee #1:

The authors address a relevant physiological question and provide new evidence that resveratrol supplementation diminishes the negative effects of a high-fat diet on muscle contractile function in mice. However, I believe manuscript requires some major and minor corrections.

Major comments:

- 1) The manuscript only references studies that support resveratrol as a potential treatment for obesity-induced skeletal muscle impairments (lines 99-108). However, this topic remains controversial. While animal studies suggest that resveratrol may improve skeletal muscle structure, function, and performance, clinical trials have yet to provide definitive evidence (Poulsen MM et al., Diabetes 2013; Toniolo L et al., Nutrients 2023). This contradicts what is mentioned in the manuscript (lines 107 and 108).**

Response: The authors appreciate this comment and acknowledge the need to highlight that there are studies which have found resveratrol had limited/no anti-obesogenic effects, specifically relating to markers of SkM health. As such, additional text has been added (Line 110-126) to highlight the conflicting results (Poulsen MM et al., 2013 Diabetes; van der Made et al., 2015 PLOS one) and identify that in human models the anti-obesogenic effects of RES appear influenced by a variety of factors such as dose/duration dependent modes of action, trial design, homogeneity of participants used and magnitude of impaired cell metabolism.

- 2) Please indicate in the introduction whether previous studies include or lack in vivo, in situ, or in vitro functional experiments, along with the advantages and disadvantages of each approach. This will help clarify the knowledge gap the study aims to address. E.g. previous experiments show that resveratrol improves muscle strength (i.e., grip strength) in rats with high-fat diet-induced obesity (Huang Y et al., Aging (Albany NY) 2019). This is an important finding that is not mentioned in the manuscript and although muscle strength and muscle contractile function are distinct concepts, they are related, and including this will help clarify the novelty of the paper. Some clear rationale is also needed in the abstract (lines 30-34).**

Response: The authors thank the reviewer for this comment and agree that it is important to highlight the advantages and disadvantages of different models to assess skeletal muscle function. As such we have provided additional information of in vivo, in situ and in vitro approaches and models which we believe highlights the advantages and disadvantages of each model, without detracting from the main purpose of the manuscript. [Please see line 141 – 177]

- 3) Potential mechanisms for resveratrol benefits in skeletal muscle are mentioned in the manuscript (lines 102-107, 112-114, 128-130), but these are poorly described. Please provide a more detailed description of these mechanisms, particularly those related to Sirtuin1. This will help readers to understand the rationale for measuring Sirtuin1. It is also important to show the controversies regarding Sirtuin1 activation by resveratrol. Resveratrol is not a direct Sirt1 activator, and attributing its effects in high fat-fed mice to Sirt1 without genetic validation has been problematic (Brenner C, Life Metab 2022).

Response: Thank you for the feedback on these sections. We have taken on board this comment and attempted to provide a more detailed section on the potential mechanisms of resveratrol on skeletal muscle health and how previous literature in rodent models relates to our work (Lines 102-108, Lines 111-113 and Lines 133-134).

- 4) The study suggests mechanisms for resveratrol's benefits in muscle but only includes functional experiments and Sirtuin1 protein measurements. Additional molecular (e.g., gene expression and/or protein levels of Sirtuin1-related targets or other pathways) and histological (e.g., fibre typing) analyses would provide mechanistic insights and strengthen the paper.

RESPONSE: The authors would like to thank the reviewer for this comment and we agree that additional mechanistic analysis would improve the paper. As such, we conducted additional protein analysis on the remaining tissue samples that provides insight into the molecular responses to resveratrol in relation to the potential induction of mitochondrial biogenesis. We investigated AMPK and ACC (down stream protein of AMPK). However, due to limited sample availability, adequate optimisation of the assays was not possible. Whilst we conducted this analysis (please see figure below) we have opted not to include it in the manuscript given further optimisation would be needed in order to give us confidence that what we have observed is a true response in these proteins. With respect to histological analysis, these experiments were not prepared for conducting this type of analysis. We have considered this in our ongoing work investigating sex specific effects of HFD with/without resveratrol, which will shed light into the effects of our experimental model on skeletal muscle morphology and phenotype. We now recognise this limitation in our work and suggest this as an area of future direction (please see line 809 -812). We have, however, included new novel data in the form of contractile assessment of the passive properties of the muscle's studied which does provide further mechanistic insight (Please see new figure 4 and lines 310-312, 495-501, 658-651 in the discussion).

Figure 1. The effect of 12 weeks high-fat diet and resveratrol on the ratio of PhosphoACC/TotalACC of isolated mouse soleus (A) and extensor digitorum longus (B). For each muscle and treatment group: n=6. Data presented as individual data points and mean \pm SD.

- 5) Outcomes of studies on the impact of obesity on muscle contractile function have been inconsistent (Tallis J et al., J Exp Biol 2018), and this is either not mentioned or poorly discussed in the current manuscript. I suggest a more detailed discussion in the section 'High-fat Diet Effects on Isolated Skeletal Muscle Function,' considering factors such as the muscle phenotype, contractility mode, feeding period and nutrient profile of diets, as these can influence the obesity response.

Response: We thank the reviewer for this feedback. As the reviewer rightly states the effects of HFD on skeletal muscle function are often complex and direct comparisons between studies can be challenging due to the impact of varying methodological approaches e.g., differing doses, diet composition, durations, age etc... (as mentioned in the introduction line 83-84). In this instance we opted to highlight where the present data follows the general trend of the literature indicating the effects of HFD are more substantial in fast-twitch EDL as the focus of the paper was on the impact of Resveratrol on contractile mechanics. However, upon consideration of the comment and review of the manuscript the authors agree that it is important to highlight the complexity of HFD responses. As such further discussion has now been included (Please see line 638-670 and 709-713). Furthermore, based on a suggestion from reviewer 2 we have also included new novel data regarding HFD

and RES effects on passive work and its contribution to active work, which has now been included in both the discussion and results.

Minor comments:

1) Please define obesity (e.g., increased body weight or increased adiposity), since there is evidence that increased body weight can increase muscle force and power in postural muscles (lines 65).

Response: Obesity has been defined as per reviewer suggestion (excessive accumulation of adipose Line 65). The authors acknowledge that there is some evidence for increased force and torque in postural muscles in individuals living with obesity. As such, the wording in the latter part of the sentence has been rephrased to “altered SkM contractile mechanics” to encompass changes in function beyond absolute changes in force and power.

2) Experiments were performed in female rats only such that it remains unclear whether similar findings would be observed in males. Please explain why only female mice were included or mention this in 'limitations.'

Response: Previous work examining HFD effects on isolated SkM function are specific to a single sex and we have extensive data for HFD effects on this sex, age and strain and was therefore considered an appropriate model as a starting point for this work. However, we recognise the need and importance of future work to directly compare the sex-specific effects of HFD and RES consumption on isolated skeletal muscle contractility, which has been added in the limitations and future direction (Line 812-816)

3) Not all adverse effects of high-fat diet were reversed by resveratrol - LSHR time was longer in HFD groups compared to SLD, but similar in HFD groups with or without resveratrol. Please correct this (line 42).

Response: wording has been changed as per reviewer comment (please see line 42)

4) The data mentioned in lines 180-183 should be included in the introduction to help clarify the knowledge gap between in vivo (muscle strength) and in vitro (contractile function) assessments, addressed by this study (see Major comment 2).

Response: As per major comment 2 this information has now been included.

5) Please correct Table 1. Not all significant values are marked in bold.

Response: This has been amended accordingly.

6) *I recommend simplifying the work loop power output results (lines 435-459). This can be organised into two paragraphs, one for the EDL data and one for the soleus data. I found lines 439-441 difficult to understand and recommend they be reformulated and simplified. The HFD*resveratrol interaction (lines 442-445) is the most relevant data here and should be mentioned first. The data in lines 435-437 seem unnecessary and can be excluded.*

Response: We have altered the paragraphs (Line 508-539) as per your suggestion. However, the text highlighting the effects of CF are integral and must remain as they are a key component of the analysis as it reflects the impact of length change velocity and represents a range of CF used during in vivo locomotor tasks. We have simplified as much as possible, i.e. highlighting where peak PO occurs, without providing an exhaustive lists of comparisons; which we agree would be

unnecessary and could be difficult for a reader to interpret. It is important to conduct this type of analysis as looking at only one CF may not necessarily analyse where changes in power occur and the CFs used represent a range of in vivo length change velocities.

7) The same for the Fatigue and Recovery results. This can be organised into two paragraphs - the EDL data first followed by the soleus data.

Response: This has been changed per reviewer suggestion

8) I had trouble understanding why muscle fatiguability was assessed as time to 50% fatigue. Also, lines 478 and 479 state that 'for the EDL, time to 50% fatigue was significantly lower in RES treated groups compared to control groups,' but line 664 states the opposite: 'In EDL, time to fatigue was higher in RES treated muscle compared to controls.' Please correct these inconsistencies.

Response: The literature examining the impact of HFD induced obesity on isolated skeletal muscle fatigue resistance, at present, does not have a standardised approach for measuring fatigue. In this study (and previous work from our laboratory) we opted to measure time to a 50% reduction in work loop power (relative to maximum power output) as the fatigue response is still linear at this time point. During the later stages of the fatigue protocol the fatigue response begins to plateau, or work becomes negative. Capturing the fatigue response during this plateau region would not represent a typical in vivo fatigue response and could provide misleading results. In this study we have opted to analyse a specific time point and the entire fatigue run using a statistical parametric mapping approach to provide more information on rate and time of fatigue, which could be independent of each other. For example, the time to fatigue could be the same, but the process to this standardised time point could be different.

Typographical errors have been changed as per the reviewer's suggestions (line 558-768)

9) The content in lines 548-552 appears somewhat redundant and could be simplified.

Response: This has been reduced and simplified as per reviewer suggestion (line 627-635)

Referee #2:

This ms is a report on the mechanical properties of muscles (fast-twitch extensor digitorum longus and slow-twitch soleus) isolated from mice fed a standard or high fat diet with or without Resveratrol (RES). The aim was test the hypothesis that RES would reduce the effects of the high fat diet on body fat and muscle performance, and that increased expression of SIRT1 would occur. The experiments were well designed to test these hypotheses.

The authors quite rightly point out the value of such in vitro experiments, which were ideal for discovering, for example, that the performance in the power output tests in response to diet and RES were different for the two muscle types. I'm sure this is trying to tell us something about the mechanisms of the diet and RES actions, and more discussion along these lines would have been welcome.

RESPONSE: The authors are pleased that the reviewer agrees upon the interest of our work and considers it worthy of publication. We would like to thank the reviewer for dedicating the time to provide a detailed review of our submission and the constructive feedback provided which has been useful in helping us to improve the quality of the manuscript. Below we have provided a commentary outlining the amendments that have been made and we hope we have now adequately addressed your concerns.

The points below are suggestions for improvement of the presentation.

Specific points

L88. Insert ", " after Messa et al. 2020).

Amended as per suggestion. (Line 88)

L103. Change "opposes" to "oppose".

Amended (line 110)

L176. Insert "in vivo" after "and".

Response: In vitro has been included as the studies cited measure isolated skeletal muscle mechanics (Line 216)

Giving that the solubility of CO₂ is strongly temperature dependent (more soluble at lower temperature), I'm curious about the pH of saline (same ionic composition and equilibrated with 5% CO₂).

L201. What was the pH during dissection of the at 1-3o?

L203. "... pH7.55 at room temperature before oxygenation". Before equilibration with 95% O₂, 5% CO₂, I would have expected CaCO₃ to have precipitated.

L216. What was the pH during the tests of contractility at 37oC?

Response: In our laboratory we make krebs-heinslet solution daily and have not seen CaCO₃ precipitate prior to equilibration with 95/5. As the reviewer rightly states temperature and gas flow rate do indeed alter the pH of our buffer. During oxygenated dissection at 1-3°C the pH of the buffer ranges from 7.23-7.27. During contractile assessments (37°C oxygenated) the pH of the buffer ranges from 7.4-7.42. This has now been included in the manuscript (Please see line 243-244)

L253. Were work-loops during movement without stimulation recorded to detect passive contributions of parallel elasticity (PE) to force? Were any measurements of series elasticity (SE) made? There seems to be no mention of parallel and/or series elasticity. Parallel elastic force would add to total force during movement; changes in the length of the series elasticity would affect the velocity of filament sliding during the work-loops (and thus the work done by the contractile component). When/if obesity and/or RES change work-loops, it would be useful to know how much, if any, of the changes are due to passive as opposed to contractile properties.

Response: We thank the reviewer for this comment. Information on the passive contribution to active work provides additional novel insight into potential mechanisms for changes in power output. Yes, passive work was recorded during unstimulated work loop cycles. We have processed and statistically analysed passive work produced at the control cycle frequency for both the soleus and EDL (new figure 4). There were no differences between treatment groups in passive work produced in the EDL, which could indicate that HFD-induced reductions in PO are mostly related to alterations to the active contractile properties of the tissue. Interestingly, passive work was greater in HFD soleus but this did not result in any changes in acute active PO. Additional text is included in the discussion (line 647-650 and 658-661).

L284-285. Deteriorating performance due to anoxic core etc. and correction of the net-work. Was such a correction made? If yes, state the size of the correction.

Response: Yes, a correction was made. As such, this has now been included (please see line 329-311)

L414 & Table 1. Clarify: Mass and csa are not larger for HFD groups compared to SLD groups? Any evidence about intramuscular fat?

Response: That is correct, muscle mass and CSA did not differ between HFD and SLD groups. Although this was approaching significance, with a moderate effect size, in EDL CSA (see table 1). In this instance we do not have any data on intramuscular fat. Given that histological analysis of soleus and EDL from female CD-1 mice of a similar age and duration of HFD (16-weeks high-fat forage diet) exhibit an increase intramyocellular lipids (Messa et al., 2020) we would expect to see a similar response in the present study. In the future we will endeavour to examine intramuscular fat in our HFD studies.

Messa, G.A.M., Piasecki, M., Hurst, J., Hill, C., Tallis, J., and Degens, H., 2020. The impact of a high-fat diet in mice is dependent on duration and age and differs between muscles. *The Journal of Experimental Biology*, 223 (6), jeb217117.

L430 & Fig 3, L464 & Fig 4. Indicate the rationale for presenting the results not normalized for muscle CSA or mass, respectively. I don't see the point.

Response: When reporting results on isolated muscle performance we report both absolute and normalised contractile performance as absolute data gives an indication of the maximum force and power producing capacity of the muscle, important for the present study as previous research. In some instances, the absolute data indicates that contractile performance is greater in HFD/obese populations and it is speculated that this occurs as a training adaptation to manoeuvring and controlling an elevated body mass [Tallis et al., 2017; Erskine et al., 2017; Choi et al., 2015]. Furthermore, absolute function is an important marker of locomotor performance and dynamic postural control. What absolute data fails to provide is the intrinsic force producing capacity of the muscle, indicative of changes in muscle quality. This information can be ascertained from contractile performance normalised to muscle mass or CSA.

Tallis, J., Hill, C., James, R.S., Cox, V.M., and Seebacher, F., 2017. The effect of obesity on the contractile performance of isolated mouse soleus, EDL, and diaphragm muscles. *Journal of Applied Physiology*, 122 (1), 170–181.

Choi, S.J., Files, D.C., Zhang, T., Wang, Z.-M., Messi, M.L., Gregory, H., Stone, J., Lyles, M.F., Dhar, S., Marsh, A.P., et al. 2015. Intramyocellular Lipid and Impaired Myofiber Contraction in Normal Weight and Obese Older Adults. *The Journals of Gerontology Series A: Biological Sciences and Medical Sciences* 71(4): 557–564. doi:10.1093/gerona/glv169.

Erskine, R.M., Tomlinson, D.J., Morse, C.I., Winwood, K., Hampson, P., Lord, J.M., and Onambélé, G.L. 2017. The individual and combined effects of obesity- and ageing-induced systemic inflammation on human skeletal muscle properties. *Int J Obes* 41(1): 102–111. doi:10.1038/ijo.2016.151.

L433. "* indicates significant difference at P<0.05". Difference between the group marked with * and which other group?

Response: This has been amended to clarify that it is a significant difference between HFD groups and SLD groups (line 492-493)

L436. Fig 3A should be Fig 4A.

Amended as per suggestion, but this is now 5A given the inclusion of the passive data (line 510)

L469. "* indicates significant difference at P<0.05". Difference between the group marked with * and which other group?

Response: This has been amended with additional symbols used to clarify significant difference between HFD groups and SLD groups (soleus power output normalised to body mass) and HFD only vs all other groups (all EDL power outputs) (line 545-547)

L478. Change "...lower in RES" to "...less in HFD RES treated". In Fig 6A the times look extremely similar for the SLD groups.

Response: Wording has been changed to accurately capture the outcome of the statistical test. The authors agree that the times for the SLD groups time to fatigue are extremely close and the statistical outcome is likely influenced by the HFDRES group. The statistical approach used (commonly implemented in data sets of this nature) aims to determine the effects of the two factors of diet (HFD V SLD) and treatment (RES vs CON) and where appropriate the interaction between them. Where interactions are not observed factors are considered for main effects. Whilst the effects of HFDRES on time to fatigue were not great enough to meet critical alpha for a significant interaction it would appear to influence the main effect of RES. Further discussion has been added to highlight that the statistical outcome from the multifactorial ANOVA should be interpreted with caution for the reasons described (Line 767-773). The outcomes from the statistical parametric mapping on rate of fatigue would appear to support this notion with HFDRES fatiguing at a greater rate during WL 4-14 when compared to all other groups.

L542. Fig 8A & D should be Fig 8A & B.

Amended as per suggestion, but now 9A & B (Line 621)

L 556. Delete "synergistically". "...consumed with" is enough. "Synergistically consumed" does not make sense. When there is an effect of HFD and RES on muscle performance, it seems antagonistic (HFD reduces performance, and RES counteracts) rather than synergistic.

Amended as per suggestion (line 632)

L563. Remove "to suggest".

Amended as per suggestion (Line 638)

L566. I think you mean "consistent" of "in agreement", rather than "concurrent" which means occurring at the same time.

Text has changed to include a more detailed overview of HFD effects on isolated skeletal muscle function as per suggestions from reviewer 1 (line 638-651)

L593. How do you know the force is "passive" rather than active (due to force produced by attached crossbridges)?

Response: We thank the reviewer for their comment, and they raise a valid point. It was not our intention to contribute the negative work to the passive elements within the tissue, rather the force being produced whilst the muscle is not being stimulated. As such, we have amended the sentence for clarity [please see line 691-692]

L634. See also L414 & Table 1. Do you have any evidence about intramuscular lipid accumulation?

Response: We thank the reviewer for this comment. In this instance we do not have direct evidence of intramuscular lipid accumulation in our tissue. However, based on our previous published work using a similar methodological approach, we would expect an increase in intramuscular lipid in HFD treated muscle. Based on the reviewers comments we will examine intramuscular lipids within the tissue in our future HFD studies. This has also been added to the limitations and future directions (Line 809-812)

Messa, G.A.M., Piasecki, M., Hurst, J., Hill, C., Tallis, J., and Degens, H., 2020. The impact of a high-fat diet in mice is dependent on duration and age, and differs between muscles. *The Journal of Experimental Biology*, 223 (6), jeb217117.

L664. Would be clearer to say that fatigue was greater, rather than higher.

Response: Wording has been altered for clarity (line 768).

L703 &704. Clarify what "not concurrent" means here.

Response: Wording has been changed to aid with clarity. What we are conveying is where changes in concentric muscle quality occur it does not mean a change in eccentric muscle quality will also happen and vice versa (please see line 820-821).

END OF COMMENTS

Dear Dr Shelley,

Re: JP-RP-2025-287056R1 "Adverse Effects of High-Fat Diet Consumption on Contractile Mechanics of Isolated Mouse Skeletal Muscle are Reduced When Supplemented with Resveratrol" by Sharn Shelley, Rob S James, Steven J Eustace, Mark C Turner, Ryan Brett, Emma Eyre, and Jason Tallis

Thank you for submitting your manuscript to The Journal of Physiology. It has been assessed by a Reviewing Editor and by 2 expert referees and we are pleased to tell you that it is acceptable for publication following satisfactory revision.

REVISION CHECKLIST:

We look forward to receiving your revised submission.

Yours sincerely,

Karyn Hamilton
Senior Editor
The Journal of Physiology

REQUIRED ITEMS

- Papers must comply with the Statistics Policy: https://jp.msubmit.net/cgi-bin/main.plex?form_type=display_requirements#statistics.

In summary:

- If n {less than or equal to} 30, all data points must be plotted in the figure in a way that reveals their range and distribution. A bar graph with data points overlaid, a box and whisker plot or a violin plot (preferably with data points included) are acceptable formats.
- If $n > 30$, then the entire raw dataset must be made available either as supporting information, or hosted on a not-for-profit repository, e.g. FigShare, with access details provided in the manuscript.
- 'n' clearly defined (e.g. x cells from y slices in z animals) in the Methods. Authors should be mindful of pseudoreplication.
- All relevant 'n' values must be clearly stated in the main text, figures and tables.
- The most appropriate summary statistic (e.g. mean or median and standard deviation) must be used. Standard Error of the Mean (SEM) alone is not permitted.
- Exact p values must be stated. Authors must not use 'greater than' or 'less than'. Exact p values must be stated to three significant figures even when 'no statistical significance' is claimed.

EDITOR COMMENTS

Reviewing Editor:

Dear Dr. Shelley and Colleagues, I am please to report that the Reviewers had positive remarks on your revision. There are additional minor critiques on the paper aimed at improving clarity. Please see suggestions from Reviewer 1.

Please also see 'Required Items' above.

Senior Editor:

Comments for Authors to ensure the paper complies with the Statistics Policy (Required):

Thank you for visiting the statistics policy. Could you please go through the manuscript one more time to clarify p-values? In most places you provide precise p-values in accordance with the policy. But in a number of cases you indicate, for example, $p < 0.014$ or $p > 0.165$. Are you simply meaning to say $p = 0.014$ or $p = 0.165$? If not, I'm not clear what your nomenclature means.

Comments to the Author:

Thank you for submitting your revised manuscript along with responses to your peer reviews. For the most part, the

Referees are pleased with the revisions and indicate that the revised manuscript is more impactful as a result. They do still have a number of minor concerns that we would like to invite you to address with another set of responses and revisions. At this point, we would like to Provisionally Accept your manuscript pending satisfactory revisions to address these remaining concerns. Thank you for your continued interest in The Journal and we look forward to seeing your revised work!

REFEREE COMMENTS

Referee #1:

Comments for the Authors are attached. [see file attached to this email]

Referee #2:

The authors have dealt with the points I raised in my first review in a satisfactory way.

Additional minor points

L261. "and" should be "at".

L 347 "expect" should be "except"

L462, 468. I find "treated" in the phrases such as "HFD treated" & "SLD treated" confusing; I initially took it to mean HFD and treated with RES, etc. I'd suggest you simply refer to the groups as HFD, HFD RES, SLD and SLD RES, in line with labelling of Figs and Tables.

L505 remove "for" from phrase "produced greater for passive"

END OF COMMENTS

The authors have addressed most of my previous comments. However, the manuscript still requires further simplification and clarity. For example:

The introduction is quite extensive (eight paragraphs) and could be more concise by reducing redundancy and restructuring it into fewer paragraphs. E.g., in the first paragraph (lines 72-76), it is stated that *supplementation of nutraceuticals is gaining substantial attention as growing evidence...*, yet the second paragraph shifts to discussing skeletal muscle properties, *in vivo* vs. *in vitro* assessments, and the impact of HFD on muscle contractile properties. The third paragraph then repeats that *Nutraceuticals have garnered considerable attention in recent years...* This affects the manuscript's readability and could be simplified as:

First paragraph – Impact of obesity on skeletal muscle function and contractile properties (there is no need to include details that fall outside the paper's scope).

Second paragraph – Effects of resveratrol on skeletal muscle function and contractile properties in individuals and animals with obesity, including possible mechanisms, especially those related to Sirtuin1.

Third paragraph – Knowledge gap that the study aims to address, specifying whether previous studies included or lacked *in vivo*, *in situ*, or *in vitro* functional experiments, along with the pros and cons of each approach. The authors have already addressed this, but further clarification is still needed.

The manuscript includes redundant sections and simplifying them would improve its clarity. Below is a clear example that could be simplified.

The present data adds further evidence indicating that HFD consumption adversely affects fast twitch EDL muscle mechanics, whereby relaxation kinetics (Bott et al. 2017, Shelley et al. 2023), absolute power production (Shelley et al. 2023, Tallis et al. 2024) and muscle quality (Tallis et al. 2017, 2024, Hurst et al. 2019, Shelley et al. 2023) are all diminished, **but isometric force and stress is unchanged (Ciapaite et al. 2015, Bott et al. 2017, Hurst et al. 2019, Shelley et al. 2023).** Whilst the present data suggests HFD consumption effects have little effect on EDL isometric force and stress, this is not always the case in studies utilising feeding durations ≥ 12 weeks (Eshima et al. 2017, 2020, Tallis et al. 2017, Shelley et al. 2023).

The authors have addressed my previous comment (5) by expanding the discussion in the section **“High-fat Diet Effects on Isolated Skeletal Muscle Function,”** taking into account factors like muscle phenotype, contractility mode, feeding period, and diet nutrient profile. To improve readability, I recommend repositioning the last paragraph (highlighted in red below) to the beginning of the section.

It should be noted that the effects of HFD consumption on contractile performance of isolated SkM and the associated mechanisms of action have shown to be influenced by study design heterogeneity (e.g., sex, age, strain of rodent, muscle phenotype, contractile mode, feeding duration, dietary composition, and experimental test temperature), often making direct comparisons between studies difficult (Tallis et al. 2018, 2022).

Response to Reviewer Comments

Senior Editor

Comments to the Author: Thank you for submitting your revised manuscript along with responses to your peer reviews. For the most part, the Referees are pleased with the revisions and indicate that the revised manuscript is more impactful as a result. They do still have a number of minor concerns that we would like to invite you to address with another set of responses and revisions. At this point, we would like to Provisionally Accept your manuscript pending satisfactory revisions to address these remaining concerns. Thank you for your continued interest in The Journal and we look forward to seeing your revised work!

Response: The authors would like to thank the editor and the independent reviewers for the opportunity to revise our submission. The reviews provided have been rigorous and constructive and have allowed us to improve the quality of our manuscript. We are pleased that the work is now considered worthy of publication, subject to completing the suggested minor amendments.

Comments for Authors to ensure the paper complies with the Statistics Policy (Required): Thank you for visiting the statistics policy. Could you please go through the manuscript one more time to clarify p-values? In most places you provide precise p-values in accordance with the policy. But in a number of cases you indicate, for example, $p < 0.014$ or $p > 0.165$. Are you simply meaning to say $p = 0.014$ or $p = 0.165$? If not, I'm not clear what your nomenclature means.

Response: We have added exact p-values for all the statistical analysis conducted on each of the figures. However, in some places in the results text, we have summarised the exact p-values in the figures to keep the results section concise. Given the volume of statistical analysis conducted, citing every p-value in text would make the written results difficult to read, and as such, we opted to present the exact p-values in the figures.

The only exception is where pairwise comparison for cycle frequency are reported. Reporting exact p-values for each cycle frequency would require a further 126 p-value citations. We are again worried that this would influence the readability of the results section. Furthermore, given the lack of treatment* cycle frequency interactions, pairwise comparisons for cycle frequency have limited importance to the primary research question. We have now outlined this approach on lines 383-385 of the manuscript. However, if this approach is not suitable we could present post hoc test data for cycle frequency comparisons in a supplementary table?

Reviewer 1

Response: We are pleased that we were able to address most of your comments in the previous submission and thank the reviewer for again taking the time to consider our manuscript and for the constructive feedback that is valuable to improving the quality of our submission. We have worked to improve the clarity of the communication as per the points outlined below and provide specific commentary to outline the actions that have been taken.

The authors have addressed most of my previous comments. However, the manuscript still requires further simplification and clarity. For example: The introduction is quite extensive (eight paragraphs) and could be more concise by reducing redundancy and restructuring it into fewer paragraphs. E.g., in the first paragraph (lines 72-76), it is stated that

supplementation of nutraceuticals is gaining substantial attention as growing evidence..., yet the second paragraph shifts to discussing skeletal muscle properties, in vivo vs. in vitro assessments, and the impact of HFD on muscle contractile properties. The third paragraph then repeats that Nutraceuticals have garnered considerable attention in recent years... This affects the manuscript's readability and could be simplified as:

First paragraph – Impact of obesity on skeletal muscle function and contractile properties (there is no need to include details that fall outside the paper's scope).

Second paragraph – Effects of resveratrol on skeletal muscle function and contractile properties in individuals and animals with obesity, including possible mechanisms, especially those related to Sirtuin1.

Third paragraph – Knowledge gap that the study aims to address, specifying whether previous studies included or lacked in vivo, in situ, or in vitro functional experiments, along with the pros and cons of each approach. The authors have already addressed this, but further clarification is still needed.

Response: We agree that the combination of the original structure of the introduction and the addition to this section following the first review have made the section a little unwieldy which may influence readability. As such, we have reworked the introduction to increase clarity in the line of argument, reduced repetition and focused on the content directly relevant to the question posed. Whilst we closely followed the model suggested, we were unable to condense this section to three paragraphs and keep the suggested level of detail. However, the number of paragraphs has been reduced and the length of the introduction reduced by ~400 words. We hope the reviewer agrees that the introduction is now better balanced with respect to detail and readability.

The manuscript includes redundant sections and simplifying them would improve its clarity. Below is a clear example that could be simplified.

The present data adds further evidence indicating that HFD consumption adversely affects fast twitch EDL muscle mechanics, whereby relaxation kinetics (Bott et al. 2017, Shelley et al. 2023), absolute power production (Shelley et al. 2023, Tallis et al. 2024) and muscle quality (Tallis et al. 2017, 2024, Hurst et al. 2019, Shelley et al. 2023) **are all diminished, but isometric force and stress is unchanged** (Ciapaite et al. 2015, Bott et al. 2017, Hurst et al. 2019, Shelley et al. 2023). Whilst the present data suggests HFD consumption effects have little effect on EDL isometric force and stress, this is not always the case in studies utilising feeding durations ≥ 12 weeks (Eshima et al. 2017, 2020, Tallis et al. 2017, Shelley et al. 2023).

Response: As suggested, we have revised the discussion, including the section suggested, to remove redundant text. We hope this improves clarity.

The authors have addressed my previous comment (5) by expanding the discussion in the section "**High-fat Diet Effects on Isolated Skeletal Muscle Function**," taking into account factors like muscle phenotype, contractility mode, feeding period, and diet nutrient profile. To improve readability, I recommend repositioning the last paragraph (highlighted in red below) to the beginning of the section.

It should be noted that the effects of HFD consumption on contractile performance of isolated SKM and the associated mechanisms of action have shown to be influenced by study design heterogeneity (e.g., sex, age, strain of rodent, muscle phenotype, contractile

mode, feeding duration, dietary composition, and experimental test temperature), often making direct comparisons between studies difficult (Tallis et al. 2018, 2022)

Response: We agree with this suggestion and have amended as advised. This text can now be found on lines 602-606.

Reviewer 2

The authors have dealt with the points I raised in my first review in a satisfactory way.

Response: We are pleased that we have adequately amended the manuscript based on your suggestions and that you now consider our work worthy of publication. We thank the reviewer for helping to improve the quality of our manuscript, and for the further minor suggestions provided. We have now addressed these points in the updated version of the submission.

Additional minor points

L261. "and" should be "at".

Response: Amended as suggested (line 229).

L 347 "expect" should be "except"

Response: Amended as suggested (line 314).

L462, 468. I find "treated" in the phrases such as "HFD treated" & "SLD treated" confusing; I initially took it to mean HFD and treated with RES, etc. I'd suggest you simply refer to the groups as HFD, HFD RES, SLD and SLD RES, in line with labelling of Figs and Tables.

Response: We agree and understand why this phrasing could be misleading. As suggested, we have removed references to "treated" here and in other relevant parts of the manuscript.

L505 remove "for" from phrase "produced greater for passive"

Response: Amended as suggested (line 470).

Dear Dr Tallis,

Re: JP-RP-2025-287056R2 "Adverse Effects of High-Fat Diet Consumption on Contractile Mechanics of Isolated Mouse Skeletal Muscle are Reduced When Supplemented with Resveratrol" by Sharn Shelley, Rob S James, Steven J Eustace, Mark C Turner, Ryan Brett, Emma Eyre, and Jason Tallis

We are pleased to tell you that your paper has been accepted for publication in The Journal of Physiology.

Yours sincerely,

Karyn Hamilton
Senior Editor
The Journal of Physiology

If you would like to receive our 'Research Roundup', a monthly newsletter highlighting the cutting-edge research published in The Physiological Society's family of journals (The Journal of Physiology, Experimental Physiology, Physiological Reports, The Journal of Nutritional Physiology and The Journal of Precision Medicine: Health and Disease), please click this link, fill in your name and email address and select 'Research Roundup':
<https://www.physoc.org/journals-and-media/membernews>

- **TRANSPARENT PEER REVIEW POLICY:** To improve the transparency of its peer review process, The Journal of Physiology publishes online as supporting information the peer review history of all articles accepted for publication. Readers will have access to decision letters, including Editors' comments and referee reports, for each version of the manuscript as well as any author responses to peer review comments. Referees can decide whether or not they wish to be named on the peer review history document.
- You can help your research get the attention it deserves! Check out Wiley's free Promotion Guide for best-practice recommendations for promoting your work at: www.wileyauthors.com/eoo/guide. You can learn more about Wiley Editing Services which offers professional video, design, and writing services to create shareable video abstracts, infographics, conference posters, lay summaries, and research news stories for your research at: www.wileyauthors.com/eoo/promotion.
- **IMPORTANT NOTICE ABOUT OPEN ACCESS:** To assist authors whose funding agencies mandate public access to published research findings sooner than 12 months after publication, The Journal of Physiology allows authors to pay an Open Access (OA) fee to have their papers made freely available immediately on publication.

EDITOR COMMENTS

Reviewing Editor:

Dear Dr. Tallis and colleagues, thank you for your efforts in modifying your manuscript. These modifications have improved the paper. I am pleased to recommend your article for acceptance.

Senior Editor:

Thank you for your careful manuscript revisions. We are pleased to accept it for publication in The Journal of Physiology. Congratulations and thank you for your interest in The Journal!